# Exogenous Proline Optimizes Osmotic Adjustment Substances and Active Oxygen Metabolism of Maize Embryo under Low-Temperature Stress and Metabolomic Analysis

**Shiyu Zuo [1,2], Yuetao Zuo [1,2], Wanrong Gu [1,2], Shi Wei [1,2] and Jing Li [1,2,*]**

[1] College of Agriculture, Northeast Agricultural University, Harbin 150030, China; 1603019@neau.edu.cn (S.Z.); s190301015@neau.edu.cn (Y.Z.); wanronggu@neau.edu.cn (W.G.); maizelab@neau.edu.cn (S.W.)

[2] Key Laboratory of Germplasm Innovation and Physiological Ecology of Grain Crops in China Cold Regions, Ministry of Education, Harbin 150030, China

[*] Correspondence: jingli1027@neau.edu.cn; Tel.: +86-451-551-90304

**Abstract:** Maize (*Zea mays* L.) is more sensitive to low-temperature stress in the early growth period. The study was to explore the response mechanism of proline to low-temperature stress during maize seed germination. Maize varieties Xinxin 2 (low-temperature insensitive) and Damin 3307 (low-temperature sensitive) were chosen as the test materials, setting the normal temperature for germination (22 °C/10 °C, 9d), low-temperature germination (4 °C/4 °C, 5d) and normal temperature recovery (22 °C/10 °C, 4d), combined with proline (15 mmol·L$^{-1}$) soaking treatment, to study its effects on the osmotic regulation system and antioxidant protection system of maize embryos. Metabolomics analysis was carried out to initially reveal the basis of the metabolic regulation mechanism. The results showed that the activities of superoxide dismutase (SOD), peroxidase (POD), ascorbic acid peroxidase (APX) and glutathione reductase (GR) were induced to some extent under low-temperature stress. The activities of SOD, POD, APX and GR were further enhanced in the soaking seeds with proline. Proline treatment improved the activities of catalase (CAT), monodehydrated ascorbic acid reductase (MDHAR) and dehydroascorbic acid (DHAR), increased the contents of ascorbic acid (AsA) and glutathione (GSH) and decreased the contents of oxidized ascorbic acid (DHA) and reduced glutathione (GSSG) under low-temperature stress. The ratio of AsA/DHA and GSH/GSSG increased. The increase in antioxidant enzyme activity and the content of antioxidants can help to maintain the stability of the AsA-GSH cycle, and effectively reduce the production rate of superoxide anion (O$_2^{\bullet-}$), hydrogen peroxide (H$_2$O$_2$) and malondialdehyde (MDA). Based on the UPLC-MS/MS detection platform and self-built database, 589 metabolites were detected in each treated maize embryo; 262 differential metabolites were obtained, including 32 organic acids, 28 amino acids, 20 nucleotides and their derivatives, 26 sugars and alcohols, 46 lipids, 51 alkaloids, 44 phenols and 15 other metabolites. Sixty-eight metabolic pathways involving different metabolites were obtained by KEGG enrichment analysis. The results showed that proline increased the accumulation of sorbitol, planteose, erythritose 4-phosphate, arabinose and other saccharides and alcohols in response to low-temperature stress, increased the content of osmoregulation substances under low-temperature stress. Proline also restored the TCA cycle by increasing the content of α-ketoglutarate and fumaric acid. Proline increased the contents of some amino acids (ornithine, proline, glycine, etc.), alkaloids (cocamidopropyl betaine, vanillylamine, 6-hydroxynicotinic acid, etc.), phenols (phenolic ayapin, chlorogenic acid, etc.) and vitamins (ascorbic acid, etc.) in the embryo under low-temperature stress. Combined with pathway enrichment analysis, proline could enhance the low-temperature stress resistance of germinated maize embryos by enhancing starch and sucrose metabolism, arginine and proline metabolism, biosynthesis of secondary metabolites, flavonoid biosynthesis and pentose phosphate pathway.

**Keywords:** maize embryo; low-temperature stress; proline; antioxidant system; metabonomics analysis

## 1. Introduction

Seed germination is susceptible to mechanical damage, disease and environmental stress. Germination starts from imbibition, goes through germinating stage and ends with radicle breaking through surrounding structures [1]. This complex process involves the restart of intracellular metabolic processes, such as cell water absorption and elongation, cell membrane transformation from gel phase to liquid crystal phase, respiration generation of ATP, DNA and protein repair, many new gene transcriptions and protein synthesis, etc. [2]. At the stage of seed germination, the cell structure, especially the membrane structure, is not perfect and the cell metabolism is unstable and susceptible to low temperature. Therefore, the seed germination stage is the low-temperature sensitive stage [3]. Low-temperature stress during seed germination would hinder cell elongation, reduce germination potential, germination rate, germination index, vigor index, bud length, root length, dry fresh weight of bud, and prolong germination duration [4]. Each plant has a set of optimum temperatures for its normal growth and development. Maize is a warm-loving crop, and the optimal temperature for germination is 25–35 °C. Too low temperature or long-term insufficient temperature will lead to a lower seed germination rate [5,6]. The prolongation of germination time will lead to an increase of seed infection by pathogenic bacteria, the occurrence of seed rot, and seriously affect the emergence rate of crops. Low temperature at the seedling stage will not only cause leaf wilting and leaf shedding, but also reduce the rate of leaf cell division and elongation, leading to too small leaf area, which cannot support the material and energy requirements of seedling growth and may reduce the total number of leaves [7]. At the same time, the growth of the root system will be inhibited, and the biomass will be significantly reduced [8]. Low temperature can also hinder the grain filling process of maize, resulting in a decrease in grain dry weight and yield [9].

The effect of low-temperature stress on plants first occurred in the cell membrane system. In order to survive at low temperatures, plants must maintain the integrity and fluidity of cell membrane under the condition of changing temperature, and avoid the occurrence of cell membrane rupture, electrolyte leakage, ion imbalance and reactive oxygen species (ROS) accumulation [10]. The increase of unsaturated fatty acids in membrane lipids is a common stress response of plants to low temperatures, which helps to maintain membrane fluidity at low temperatures. The low-temperature resistance of plants is positively correlated with the proportion of unsaturated fatty acids in fatty acids [11]. The degree of desaturation of total phosphatidylcholine in Sorghum cells increased under low-temperature stress [12]. After low-temperature treatment, in addition to the increase in sphingolipid saturation, more unsaturated lipids are retained [13]. Fatty acid desaturation has been studied as a potential way to improve plant low-temperature tolerance. Studies confirmed that the low-temperature resistance of plants can be significantly improved after the expression or overexpression of endogenous or exogenous desaturase genes [14–16].

The damage of low-temperature stress to cells is also related to membrane lipid peroxidation and protein destruction caused by ROS accumulation. ROS produced in plant cells is the general name for oxygenated compounds with strong oxidation ability, mainly including hydrogen peroxide ($H_2O_2$), hydroxyl ($\cdot OH$) and superoxide anion radical ($O_2^{\bullet-}$) and singlet oxygen ($^1O_2$). However, excessive ROS accumulation will cause protein damage, lipid peroxidation, nucleotide degradation, and even some cell and tissue death [17]. The content of malondialdehyde, a stable end product of ROS reaction with cell macromolecules, is considered to be a marker metabolite of the degree of peroxidation damage of reaction membrane lipids. $H_2O_2$, $O_2^{\bullet-}$ and MDA content will increase significantly in leaves of maize and wheat under low-temperature stress [18]. The accumulation of ROS caused by low-temperature stress also leads to the weakening of photosynthesis, the reduction of $CO_2$ fixation and the reduction of $NADP^+$ regeneration, which finally disrupts the operation of the electron transport chain. The change of electron flow causes the excessive reduction of enzyme complexes I and III in the mitochondrial electron transport chain, which eventually leads to the leakage of electrons to $O_2$ and the production of

ROS [19]. Under low-temperature stress, the accumulation of ROS in wheat, cucumber, maize, Arabidopsis and rice began in chloroplasts and mitochondria [20,21]. Although there are only limited antioxidants between chloroplast, mitochondria and endoplasmic reticulum membrane under normal conditions, and reactive oxygen species will accumulate in these organelles under stress, the chloroplast matrix and mitochondrial matrix contain a strong antioxidant system, which can quickly remove excessive accumulated reactive oxygen species [22]. These pieces of evidence suggest that plants have evolved complex and efficient scavenging mechanisms and regulatory pathways to monitor ROS redox homeostasis and prevent excessive ROS in cells. More than 150 genes in plants encode different ROS scavenging enzymes and signal transduction, most of which are temperature responses [23]. Plants will increase the content of reactive oxygen species scavengers, improve the activity of antioxidant enzymes and generate specific proteins to improve membrane stability, so as to prevent low-temperature stress and produce excessive ROS to maintain metabolic stability [24]. The activities of antioxidant enzymes such as SOD, CAT, APX and GPX in plants have increased significantly, accompanied by significant changes in DNA methylation/demethylation mode. The increase of antioxidant enzyme activity and DNA methylation mode are effective factors to improve the low-temperature resistance of plants [25]. Overexpression of CuZnSOD and APX in chloroplasts of transgenic plants enhances the tolerance of plants to strong light and low-temperature stress [26]. The plant antioxidant system will finely regulate the antioxidant content and antioxidant enzyme activity according to different stress types, stress intensity and stress time, so as to control the potential damage or signal characteristics of ROS [27].

Osmoregulation substance is a compatible solute dissolved in intracellular fluid or surrounding fluid, which plays an important role in maintaining cell osmotic balance. The sugar accumulation induced by chilling in Petunia leaves may protect against low temperature and avoid carbohydrate consumption [28]. Proline widely exists in higher plants and is considered to be one of the most important osmoregulation substances [29]. Under low-temperature stress, plants will accumulate a large amount of proline [30]. As an osmotic regulator, proline plays a variety of roles in plant stress tolerance. Proline can not only prevent electrolyte leakage, stabilize protein structure and activity, regulate reactive oxygen species concentration, protect membrane integrity and stabilize the subcellular structure, but also maintain appropriate $NADP^+$/NADPH ratio, regulate cell pH and redox state, induce osmotic-stress-related gene expression and clear ROS [31]. Proline has been proved to enhance the activities of different enzymes by protecting the integrity of proteins. After the stress is relieved, proline can also be used as a source of energy, carbon and nitrogen to accelerate the recovery of plants [32]. Exogenous proline plays a positive role in reducing the damage caused by stress during crop growth [33].

Proline itself is an effective active oxygen scavenger, which can inactivate $^1O_2$ by quenching [34]. Free proline and polypeptide-bound proline can react with $H_2O_2$ and ·OH to produce stable free radical adducts of proline and hydroxyproline derivatives (4-hydroxyproline and 3-hydroxyproline) [35]. However, some studies have found that proline cannot quench $^1O_2$ in buffer, which leads to doubt about the ability of proline to scavenge $^1O_2$ under plant stress [36]. The combination of density functional theory and polarized continuum model (DFT/PCM) shows that proline is favorable evidence of plant ·OH scavenger under low-temperature stress, in which proline captures ·OH through dehydrogenation extraction to produce P5C, and then produces proline under the action of P5CR enzyme [37]. Many studies have proved that the application of proline improves the activity of antioxidant enzymes and the content of antioxidants in plants under stress conditions [38]. For example, under cadmium stress, proline can promote the detoxification of $^1O_2$ free radicals by increasing the SOD activity of Solanum nigrum [39]. Exogenous proline significantly enhanced the activities of APX, MDHAR and DHAR in tobacco leaves under salt stress and reduced the accumulation of ROS [40]. After seed soaking with proline, it can increase the content of ascorbic acid and glutathione in mung beans under salt stress, enhance the activities of APX, GR and CAT, and reduce the content

of $H_2O_2$ and MDA [41]. Previous studies have studied the effects of exogenous proline on the expression of proline metabolism-related genes P5CS and P5CR and genes encoding antioxidant enzymes, superoxide dismutase, ascorbate peroxidase and catalase in Rice Seedlings under salt stress. It was found that the transcription levels of P5CS and P5CR and the expression levels of antioxidant enzyme genes in rice leaves treated with proline were significantly up-regulated [42]. Proline biosynthesis is associated with the mitogen-activated protein kinase (MAPK) cascade pathway [43]. Exogenous proline will increase ROS and aggravate the P5C-proline cycle in Arabidopsis plants with P5CDH enzyme inactivation, but there is no such phenomenon in normal plants. The above results show that the oxidation process from P5C to Glu catalyzed by P5CDH plays a key role in avoiding excessive ROS production [44].

Plant metabolic processes and metabolite content are regulated by various factors in themselves and the environment. There are many kinds of metabolites in plants, including primary metabolites, such as sugars, amino acids, nucleotides, etc., and more importantly, abundant secondary metabolites, such as phenols, terpenes, alkaloids, etc. [45]. Plant secondary metabolism is the result of the interaction between plant and environment. Secondary metabolites play an important role in improving plant stress resistance. So far, there have been a large number of reports on the analysis of plant phenotypic mechanism by metabonomics, while there are relatively few studies on the effect of exogenous substances on maize under stress. Predecessors used metabonomics to find out the different metabolites of maize in different treatment groups, which provided marker metabolites for stress resistance breeding and provenance identification [46]. Of particular concern are metabolites that can function as osmoregulators, including soluble sugars, amino acids, organic acids, polyamines and lipids [47]. The changes of these metabolites are closely related to the adaptation of plants to stress. Although there is only a weak relationship between plant growth and the content of individual metabolites, a highly significant link between biomass and specific combinations of metabolites has been confirmed [48]. Based on the metabolic spectrum during the development of Arabidopsis seeds, researchers identified the main metabolic pathways related to the development stage [49]. The combination of metabonomics and quantitative genetics is the core of our understanding of plant phenotypes [50]. Through metabonomics technology, we can deeply explore the interaction between plant and environment, which provides a scientific basis for clarifying the regulation mechanism of stress or exogenous substances and stress resistance breeding. Accordingly, the adaptive change characteristics of these metabolites are an important part of determining the stress resistance of species, which makes metabonomics unique in seeking to cover all metabolites in the whole system [51].

## 2. Materials and Methods

### 2.1. Experimental Varieties

The experiment was carried out in the key laboratory of cold region grain crop variety improvement and physiological ecology of the ministry of education, Northeast Agricultural University. Based on the results of preliminary experiments, we selected maize varieties Xinxin 2 (low-temperature insensitive, XX 2) and Damin 3307 (low-temperature sensitive, DM 3307) as experimental materials. Proline was purchased from Sigma, Saint Louis, MO, USA ($C_5H_9NO_2$, CAS NO.147-85-3).

### 2.2. Experimental Design

The seeds were disinfected with 0.1% $HgCl_2$ solution for 120 s, washed with distilled water, dried with filter paper, and put into a 500 mL beaker. The seed soaking solution set in our experiment was distilled water and 15 mmol·$L^{-1}$ proline solution. We place the above seed soaking solution in a 22 °C artificial intelligence incubator (HPG-280HX) and soak the seeds for 24 h. After soaking, we take out the seeds and wash them with distilled water. The naturally dried seeds were arranged in a 15 cm culture dish with two layers of filter paper, 30 seeds per dish, and 20 mL distilled water was added to each dish. The

seeds after soaking were used in the following four treatments, including (1) control group (CK), which germinated at 22 °C/10 °C for 9 days after soaking in distilled water (the 0th day); (2) seed soaking with proline (P), 15 mmol·L$^{-1}$ proline solution always germinated at 22 °C/10 °C for 9 days; (3) low-temperature stress treatment (L), after soaking seeds in distilled water, the seeds germinated at 4 °C/4 °C for 5 days, and then transferred to 22 °C/10 °C for 4 days; (4) low-temperature stress and proline combined treatment (L + P), after soaking seeds in 15 mmol·L$^{-1}$ proline solution, the seeds germinated at 4 °C/4 °C for 5 days, and then transferred to 22 °C/10 °C for 4 days. All seeds germinated under dark and humid conditions and each treatment was repeated 3 times. We took appropriate amounts of seeds on the 0, 1, 3, 5, 7 and 9 days of each treatment, and carefully peeled the complete seed embryo with a scalpel for determination.

### 2.3. Measurement and Methods

#### 2.3.1. Determination of O$_2^{\bullet-}$ Generation Rate

O$_2^{\bullet-}$ generation rate was determined according to the improved method of Esim and Atici [52]. A 0.5 g sample of the embryo was ground into homogenate with 2 mL of 65 mM phosphate buffer (pH 7.8). The above homogenate was prepared at 5000× *g* centrifugation for 10 min at 4 °C. Then, 1 mL supernatant was mixed with 0.9 mL of 65 mM phosphate buffer (pH 7.8) and 0.1 mL of 10 mM hydroxylamine hydrochloride, and the mixture was placed at 25 °C for 20 min. Then, 1 mL of the mixture, 1 mL of 17 mM anhydrous p-aminobenzene sulfonic acid and 1 mL of 17 mM α-naphthylamine were mixed at 25 °C for 20 min. Next, 3 mL N-butanol was added to the mixture and we determined the absorbance at 530 nm. The unit of O$_2^{\bullet-}$ generation rate was nmol·min$^{-1}$·g$^{-1}$ FW.

#### 2.3.2. Determination of H$_2$O$_2$ Content

The content of H$_2$O$_2$ was determined according to the improved method of Yu et al. [53]. We took 0.5 g of embryo and ground it into homogenate with 3 mL of 50 mM phosphate buffer (pH 6.5) at 4 °C. Then the above-obtained homogenate was centrifuged at 11,500× *g* for 15 min. Next, 3 mL of supernatant and 1 mL of 0.1% TiCl$_4$ were mixed in 20% H$_2$SO$_4$ (*v/v*) and the mixture was placed at 25 °C for 10 min, and then the mixture was centrifuged again at 6000× *g* at 11 °C for 15 min. The supernatant was taken and the absorbance was measured at 410 nm. The unit of H$_2$O$_2$ content was nmol·g$^{-1}$ FW.

#### 2.3.3. Determination of MDA Content

According to the improved method of Yasar et al., thiobarbituric acid (TBA) was used as the reaction substance for determination [54]. We took 0.5 g of embryo, ground it into homogenate in 2 mL 5% trichloroacetic acid, and the homogenate was centrifuged at 11,500× *g* for 10 min. Then, 2 mL supernatant and 4 mL 0.6% thiobarbituric acid were mixed. We heated the above reaction mixture in 95 °C water for 30 min, and then quickly cooled it in an ice bath at 11,500× *g* centrifuge again for 15 min. The absorbance was measured at 450nm, 532nm and 600 nm, respectively. The unit of MDA content was μmol·g$^{-1}$ FW.

#### 2.3.4. Determination of Electrolyte Leakage (EL)

The determination of electrolyte leakage (EL) was carried out with a conductivity meter (DDS-307, Shanghai Tianda Instrument Company, Shanghai, China) according to the method of Nayyar et al. [55]. A 1 g embryo was cut into small pieces and put into a test tube. Then, 20 mL of distilled water was added to immerse the embryo. We placed the test tube on a shaking table (100 rpm) at 25 °C for 24 h and the initial conductivity (EC1) was measured with a conductivity meter. Then, the sample was bathed in 100 °C hot water for 20 min, and the conductivity (EC2) after boiling was measured at 25 °C. The calculation formula of electrolyte leakage was El = EC1/EC2 × 100%.

### 2.3.5. Determination of SOD, POD, CAT, APX, MDHAR, DHAR and GR Activities

We added liquid nitrogen to 0.5 g of embryo and ground it into a fine powder, and then added 5 mL of 100 mM potassium phosphate buffer (pH 7.4) to grind it into homogenate. Homogenate was centrifuged at $33{,}000\times g$ for 30 min, and the supernatant was collected for the determination of antioxidant enzyme activity. According to the method of Beauchamp and Fridovich, the enzyme activity was measured by measuring the inhibition of SOD on the reduction of nitroblue tetrazole (NBT) under light, and the total SOD activity was measured at 560 nm [56]. The peroxidase (POD) activity was measured with guaiacol as substrate according to the method of Hammerschmidt et al. The reaction mixture in 3 mL consists of 25 mM phosphate buffer (pH 7.0), 1.5% guaiacol, 0.4% $H_2O_2$ and 0.2 mL enzyme extract [57]. The catalase (CAT) activity was determined according to Aebi's UV absorption method. The reaction mixture contained 80 mM phosphate buffer (pH 7.0), 200 mM DTPA, 10 mM $H_2O_2$ and enzyme extract [58]. The activity of ascorbic acid peroxidase (APX) was determined according to the methods of Nakano and Asada. A 0.7 mL sample of the reaction mixture contained 50 mM potassium phosphate buffer (pH 7.0), 0.5 mM AsA, 0.1 mM $H_2O_2$, 0.1 mM EDTA and enzyme extract. APX activity was calculated by measuring the decrease of absorbance value within 1 min at 290 nm [59]. The activity of monodehydroascorbate reductase (MDHAR) was determined according to the method of Hossain et al. A 0.7 mL sample of the reaction mixture contained 50 mM Tris HCl buffer (pH 7.5), 0.2 mM NADPH, 2.5 mM AsA and enzyme extract [60]. The activity of dehydroascorbic acid reductase (DHAR) was determined with reference to Doulis's method. The reaction mixture contained 50 mM phosphate buffer (pH 7.0), 2.5 mM GSH, 0.1 mM DHA and enzyme extract [61]. Glutathione reductase (GR) activity was measured according to the method of Hasanuzzaman et al. The reaction mixture contains 0.1 mM phosphate buffer (pH 7.0), 1 mM GSSG, 1 mM EDTA-NaOH, 0.2 mM NADPH and enzyme extract [62].

### 2.3.6. Determination of AsA, DHA, GSH and GSSG Contents

A 0.5 g sample of seed embryo was put into a mortar, and 1 mL of 5% (*w/v*) precooled metaphosphoric acid extract was added to grind the seed embryo into homogenate. The homogenate was centrifuged at $11{,}500\times g$ for 15 min, and the contents of reduced ascorbic acid (AsA) and dehydroascorbic acid (DHA) were determined according to the bipyridine method of Huang et al. [63]. Then, 100 mL 0.5 M phosphate buffer (pH 7.0) and 100 mL 0.01 M dithiothreitol (DTT) were mixed to 200 mL supernatant. The reaction was allowed to stand at 25 °C for 10 min, and then 100 mL 0.5% NEM was mixed for the determination of total ascorbic acid. At the same time, we added 100 mL 0.5 m phosphate buffer (pH 7.0) and 200 mL distilled water to 200 mL supernatant for the determination of reduced ascorbic acid. We added the reaction mixture to each test tube, including 500 mL 10% TCA and 400 mL 43% $H_3PO_4$, 400 μL 4% $\alpha$-$\alpha'$-dipyridine and 200 mL 3% $FeCl_3$. After the sample was left to stand at 37 °C for 60 min, the absorbance value of red chelate in $OD_{525}$ was measured. The standard curve of AsA was used for quantification. The content of oxidized ascorbic acid (DHA) was calculated by subtracting reduced ascorbic acid (AsA) from total ascorbic acid. The contents of reduced glutathione (GSH) and oxidized glutathione (GSSG) were determined according to the method of Paradiso [64].

### 2.3.7. Determination of Soluble Protein and Soluble Sugar

The content of soluble sugar was determined according to the anthrone method described by Hansen et al. [65]. A 0.1 g sample of seed embryo was ground into homogenate in 1 mL distilled water and heated at 95 °C for 10 min. After cooling, the above-obtained homogenate was centrifuged at $8000\times g$ for 10 min and diluted to 10 mL with distilled water. A 300 μL sample of the reaction mixture contains 40 μL extract, 40 μL distilled water, 20 μL mixed reagent (1 g anthrone + 50 mL ethyl acetate) and 200 μL $H_2SO_4$ (98%). The mixture was heated at 95 °C for 10 min and the absorbance was measured at 630 nm using a microplate reader (Infinite 200 PRO). We calculated the concentration of soluble sugar

based on sucrose. The soluble protein content was measured according to the Coomassie brilliant blue G-250 method described by Bradford [66].

### 2.3.8. Metabonomic Analysis

Sample Preparation

Take 1 g of seed embryo samples from DM 3307 after 3 days of each treatment (CK, L and L + P) (the seeds in CK treatment began to break the chest, and the seeds in L and L + P treatment did not break the chest), and record them as C, L and LP, respectively. Put the sample into a freeze dryer (Scientz-100F) for vacuum freeze-drying, then grind it with a grinder (MM 400, Retsch, Haan, Germany) (30 Hz, 1.5 min) to powder, take 100 mg of powder and add 1.2 mL of 70% methanol extract, vortex once every 30 min for 30 s, a total of 6 times, and place the sample for 12 h and at $12,000 \times g$ centrifuge for 10 min, absorb the supernatant and use microporous filter membrane (0.22 μm pore size) filter the sample and store it in the injection bottle for UPLC-MS/MS analysis. Quality control samples (QC) are prepared by mixing the sample extracts and used to analyze the repeatability of samples under the same treatment method.

UPLC-MS/MS Conditions

The analytical instruments of this test include Nexera X2 ultra-high-performance liquid chromatography of Shimadzu Company, and the matched chromatographic column was Agilent SB-C18 (1.8 μm, 2.1 mm × 100 mm), the mobile phase was divided into phase A (ultrapure water, adding 0.1% formic acid) and phase B (acetonitrile, adding 0.1% formic acid). The elution gradient was divided into three stages, and the proportion of phase B was 5% at 0.00 min, respectively. The proportion of phase B increased linearly to 95% within 9.00 min and maintained at 95% for 1 min. From 10.00 to 11.10 min, the proportion of phase B decreased to 5% and balanced to 14 min with 5%. The flow rate was 0.35 mL min$^{-1}$, the column temperature is 40 °C, and the volume of injection was 4 μL.

The mass spectrometer was provided by Applied Biosystems with the model of 4500 QTRAP. The mass spectrum conditions mainly include: LIT and QQQ scanning were obtained on the triple quadrupole linear ion trap mass spectrometer (QTRAP) and AB4500 QTRAP UPLC/MS/MS system. The system was equipped with an ESI Turbo ion spray interface and can be operated in both positive and negative ion modes by Analyst 1.6.3 software (ABSciex). Positive ion mode and negative ion mode are detected separately, and the sample status and instrument hardware conditions are consistent. The operating parameters of the ESI source were as follows: ion source, turbine spray; source temperature 550 °C; ion spray voltage (IS) 5500 V (positive ion mode)/−4500 V (negative ion mode); and the ion source gas I (GSI), GAS II (GSII) and curtain GAS (CUR) were set to 50, 60 and 25.0 psi, respectively. In QQQ and LIT modes, we use 10 μMol L$^{-1}$ and 100 μMol L$^{-1}$ polypropylene glycol for instrument tuning and mass calibration. Through further DP and CE optimization, DP and CE of each MRM ion pair were completed. Based on the metabolites eluted in each period, a specific set of MRM ion pairs were monitored in each period.

Qualitative Analysis of Metabolites

We use Software Analyst 1.6.3 to process the mass spectrometry data. Based on mwdb (metal database), the metabolites in the samples were analyzed qualitatively and quantitatively by mass spectrometry. According to the second-order spectrum information, the qualitative analysis of substances was carried out, and the isotope signal was removed during the analysis. In order to compare the content difference of each metabolite in different samples among all detected metabolites, the mass spectrum peaks detected by each metabolite in different samples and quality control samples were integrated and corrected according to the information of metabolite retention time and peak type, so as to ensure the accuracy of qualitative and quantitative. The peak area of each chromatographic peak

represents the relative content of the corresponding substance. Finally, all chromatographic peak area integral data were exported and saved [67].

Metabonomic Data Analysis

We use the built-in statistical prcomp function of R software to normalize the data by unit variance scaling (UV). When analyzing metabolomic data, SIMCA (14.1) was used for principal component analysis (PCA) and orthogonal partial least squares discriminant analysis (OPLS-DA). In order to preliminarily understand the overall metabolic difference between the samples in each group and the variability between the samples in the group, and further model verification. Cluster analysis is a classified multivariate statistical analysis method, which makes individuals in the same category have as high homogeneity as possible, while between categories should have as high heterogeneity as possible. In this study, the heat map was drawn by the R software pheatmap package, and the accumulation patterns of metabolites among different samples were analyzed by hierarchical cluster analysis (HCA). The combination of fold change and VIP value of the OPLS-DA model was used to screen differential metabolites. In order to facilitate the observation of the change law of metabolites, the significantly different metabolites were normalized by unit variance scaling (UV), and the heat map was drawn by R software. Finally, the KEGG (Kyoto Encyclopedia of Genes and Genes) database was used for functional annotation and enrichment analysis of differential metabolites [68].

### 2.4. Data Analysis

According to the analysis of variance, data were statistically analyzed following standard methods using Microsoft Excel 2010 and SPSS 12.0. Differences between treatments were determined by a posteriori Tukey's test at a significance level of 0.05 and 0.01.

## 3. Results

### 3.1. MDA Contents and Electrolyte Leakage (EL)

With the extension of low-temperature stress time, the MDA content in the embryo of the two maize varieties increased gradually. On the first, third and fifth days of low-temperature stress, compared with the treatment of low-temperature stress, the MDA content in the embryo of DM 3307 increased significantly by 114.13%, 116.22% and 82.36%, respectively. The content of MDA in XX 2 embryos increased by 75.39%, 38.17% and 89.35%, respectively. Under low-temperature stress, exogenous proline treatment significantly reduced the MDA content in the embryo of two maize varieties. On the fifth day of low-temperature stress, the MDA content in the embryo of DM 3307 and XX 2 decreased by 15.21% and 7.01%, respectively. At the same time, after the low-temperature environment was removed, the MDA content of seed embryos of the two maize varieties decreased, and under low-temperature stress, seed soaking with proline could still significantly reduce the MDA content of seed embryos.

The electrolyte leakage (EL) in the embryos of two maize varieties increased significantly with the extension of low-temperature stress time. Compared with normal temperature treatment, exogenous proline can reduce the EL of seed embryos of two maize varieties, but the effect is not significant. On the fifth day of low-temperature stress, compared with normal temperature treatment, EL in DM 3307 and XX 2 embryos increased by 349.68% and 226.22%, respectively, and reached a significant level between treatments. On the fifth day of treatment, proline seed soaking treatment could significantly inhibit the increase of EL of seed embryos of two maize varieties under low-temperature stress, reducing 16.37% and 15.70%, respectively. On the fourth day of low-temperature release, the EL of seed embryos of two maize varieties decreased significantly, and exogenous proline treatment could significantly inhibit the EL of seed embryos after low-temperature treatment. The results showed that the treatment with an appropriate concentration of exogenous proline could reduce the electrolyte permeability of seed embryo and protect

the integrity of cell membrane, so as to effectively alleviate the damage of membrane lipid of maize seed embryo under low-temperature stress (Figure 1).

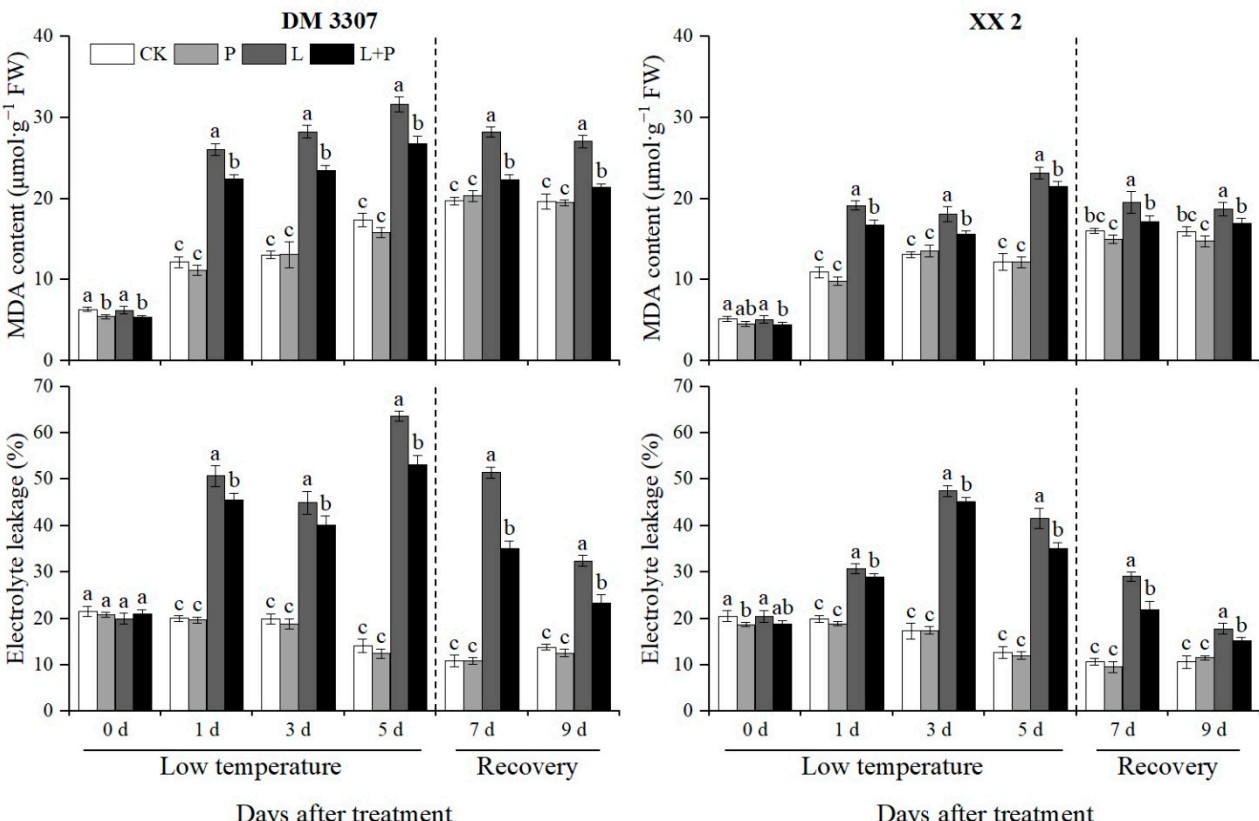

**Figure 1.** Effect of exogenous proline on MDA content and EL in embryos of maize under low-temperature stress. Notes: Data are expressed as mean ± standard deviation. Different letters within the same column indicate significant differences at the 5% level. CK: control group; P: proline treatment; L: low-temperature stress treatment; L + P: low-temperature stress + proline combined treatment. Different small letters represent significant differences between treatments ($p < 0.05$).

### 3.2. $H_2O_2$ Content and $O_2^{\bullet-}$ Generation Rate

The $H_2O_2$ content of DM 3307 and XX 2 maize embryos increased gradually with the extension of low-temperature stress time, and $O_2^{\bullet-}$ generation rate increased first and then decreased. Soaking seeds with proline at normal temperature can reduce the $H_2O_2$ content and $O_2^{\bullet-}$ generation rate of seed embryos, but there is no significance between treatments. On the fifth day of low-temperature stress, compared with normal temperature, the $H_2O_2$ content and $O_2^{\bullet-}$ generation rate of DM 3307 embryos increased by 108.77% and 76.74%, respectively, and the $H_2O_2$ content and $O_2^{\bullet-}$ generation rate of XX 2 embryo increased by 116.84% and 34.44%, respectively. Under low-temperature stress, seed soaking with proline can significantly reduce the $H_2O_2$ content and $O_2^{\bullet-}$ generation rate of maize embryos. On the fifth day of treatment, after soaking seeds with proline under low-temperature stress, the content of $H_2O_2$ content in DM 3307 and XX 2 embryos decreased by 20.41% and 7.99%, respectively, and $O_2^{\bullet-}$ generation rate decreased by 19.80% and 13.15%, respectively. The results showed that exogenous proline treatment could inhibit the production of reactive oxygen species under low-temperature stress, and the phenotype of the low-temperature sensitive variety DM 3307 was more significant. At the same time, the $H_2O_2$ content and $O_2^{\bullet-}$ generation rate decreased by varying degrees after the release of low-temperature stress, but the reactive oxygen species of maize embryos under a low-temperature environment were significantly higher than in other treatments (Figure 2).

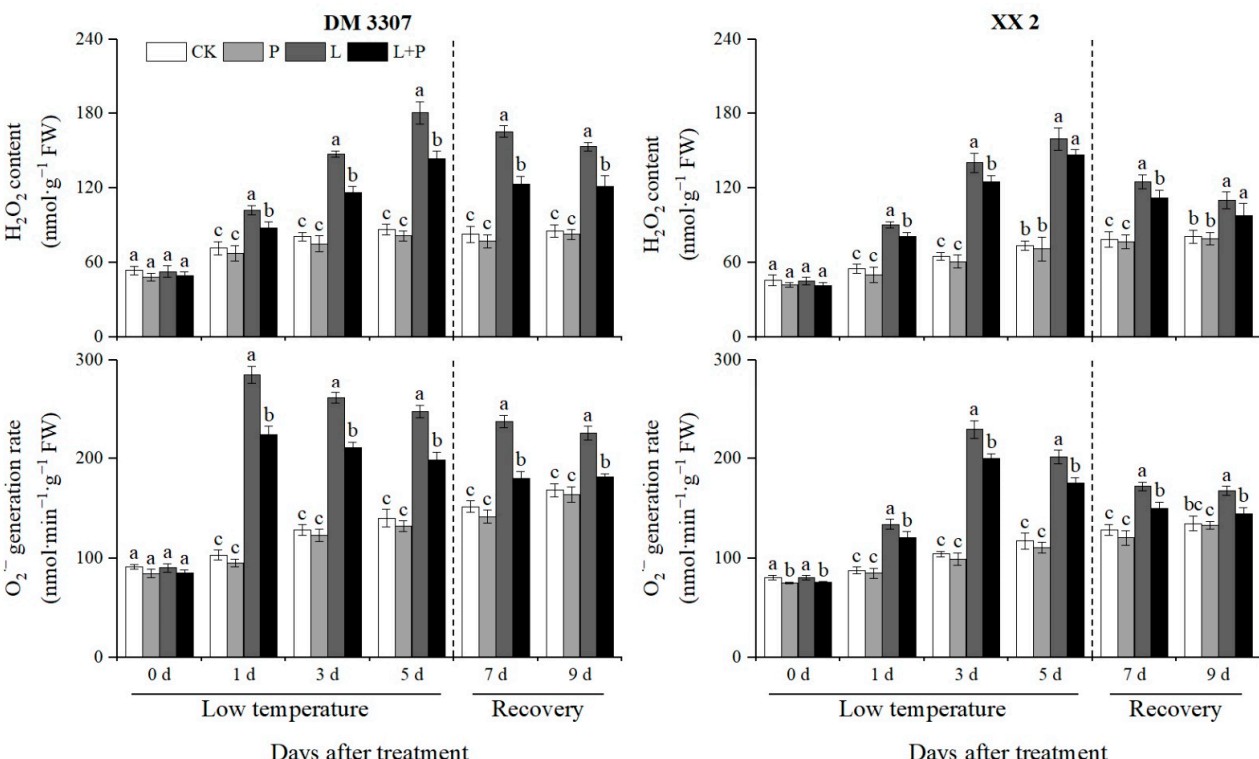

**Figure 2.** Effect of exogenous proline on $H_2O_2$ content and $O_2^{\bullet-}$ generation rate in embryos of maize under low-temperature stress. Notes: Data are expressed as mean ± standard deviation. Different letters within the same column indicate significant differences at the 5% level. CK: control group; P: proline treatment; L: low-temperature stress treatment; L + P: low-temperature stress + proline combined treatment. Different small letters represent significant differences between treatments ($p < 0.05$).

### 3.3. Activities of SOD, POD and CAT

Compared with normal temperature treatment, low-temperature stress treatment induced the increase of SOD, POD and CAT activities of DM 3307 and XX 2 maize embryos in varying degrees, and seed soaking with proline could further enhance the increase of the activities of three antioxidant enzymes. With the extension of low-temperature stress time, except for the SOD activity of XX 2 variety, it reached the highest value on the third day of low-temperature stress. On the third day of low-temperature stress, compared with the control treatment, the activities of SOD, POD and CAT in DM 3307 embryos increased by 56.03%, 98.04% and 8.03%, respectively, and the activities of SOD, POD and CAT in XX 2 embryos increased by 57.36%, 88.41% and 40.0%, respectively. On the third day of low-temperature stress treatment, proline seed soaking treatment can significantly improve the antioxidant enzyme activity of two maize varieties' embryos, so as to effectively alleviate the oxidative damage of maize embryos caused by low-temperature stress. At the same time, after the low-temperature treatment of the seed embryo was removed, the antioxidant enzyme activity of the seed embryo of XX 2 decreased more obviously than that of DM 3307, and the low-temperature treatment of seed embryo proline soaking seed can also significantly improve the antioxidant enzyme activity (Figure 3).

### 3.4. Contents of AsA, DHA and AsA/DHA Ratio

With the extension of low-temperature stress time, AsA content and AsA/DHA ratio in DM 3307 and XX 2 embryos decreased gradually, while DHA content increased gradually, reaching the extreme value on the fifth day of low-temperature stress. Compared with the normal temperature control treatment, low-temperature stress induced a significant decrease in AsA content and AsA/DHA ratio, but a significant increase in DHA content.

On the fifth day of low temperature, the AsA content and AsA/DHA ratio of DM 3307 embryos decreased by 85.44% and 90.56%, respectively, while the DHA content increased by 54.20%. The AsA content and AsA/DHA ratio of XX 2 embryo decreased by 45.84% and 61.56%, respectively, and the DHA content increased by 40.86%, indicating that DM 3307 variety is more sensitive to low-temperature stress. On the fifth day of low-temperature stress, compared with the simple low-temperature stress treatment, the AsA content of seed embryos of two maize varieties increased by 80.79% and 14.67%, respectively, while the DHA content decreased by 17.38% and 7.20%, respectively. At the same time, after removing the low-temperature condition, the AsA content and AsA/DHA of maize embryo treated with low temperature showed an increasing trend, while the DHA content showed a decreasing trend. It was also found that there was no significant difference between the AsA content of XX 2 embryo soaked in proline and the low-temperature treatment, and the AsA and DHA contents of DM 3307 embryo reached a significant level compared with the low-temperature treatment (Figure 4).

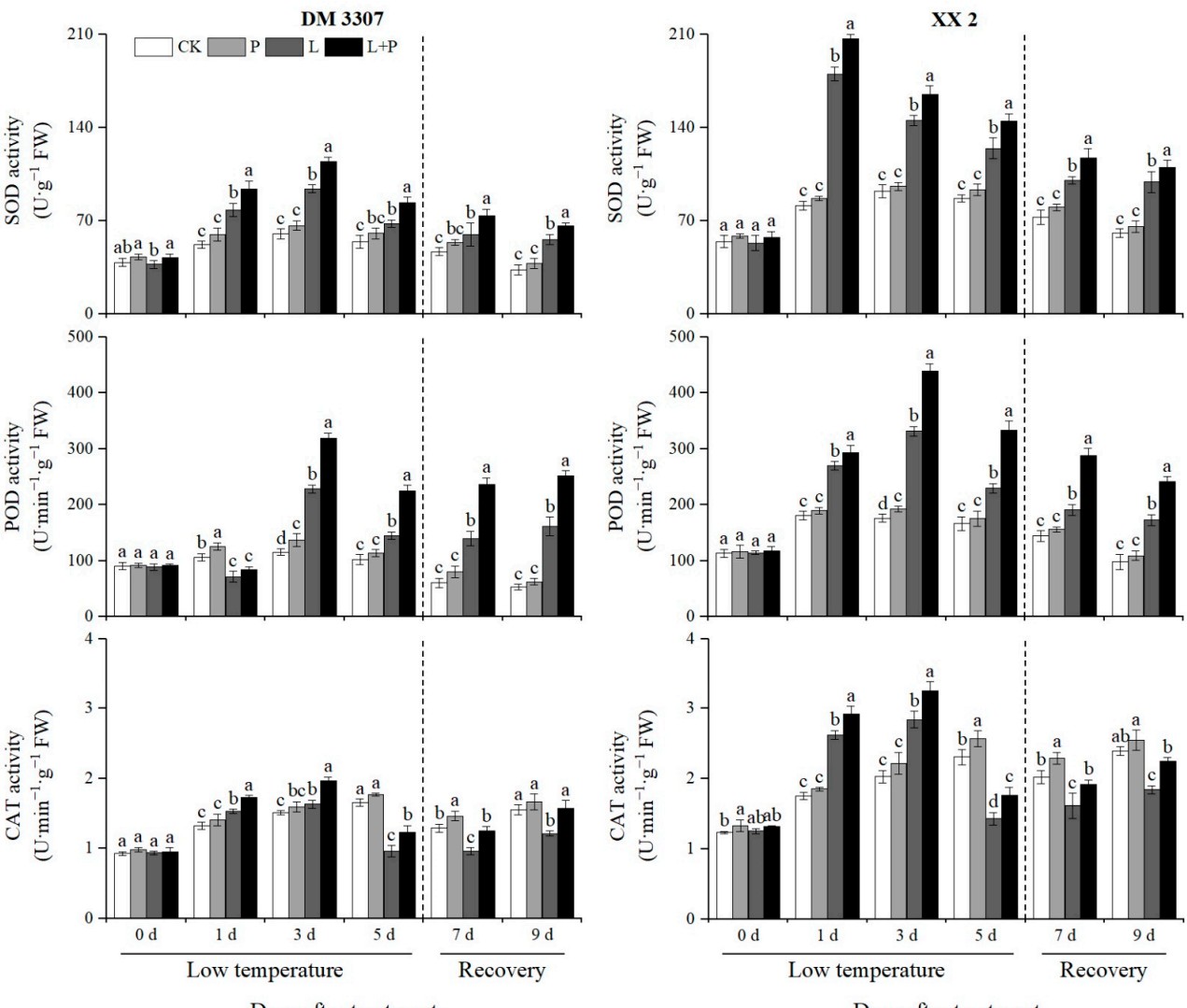

**Figure 3.** Effect of exogenous proline on the activities of SOD, POD and CAT in embryos of maize under low-temperature stress. Notes: Data are expressed as mean $\pm$ standard deviation. Different letters within the same column indicate significant differences at the 5% level. CK: control group; P: proline treatment; L: low-temperature stress treatment; L + P: low-temperature stress + proline combined treatment. Different small letters represent significant differences between treatments ($p < 0.05$).

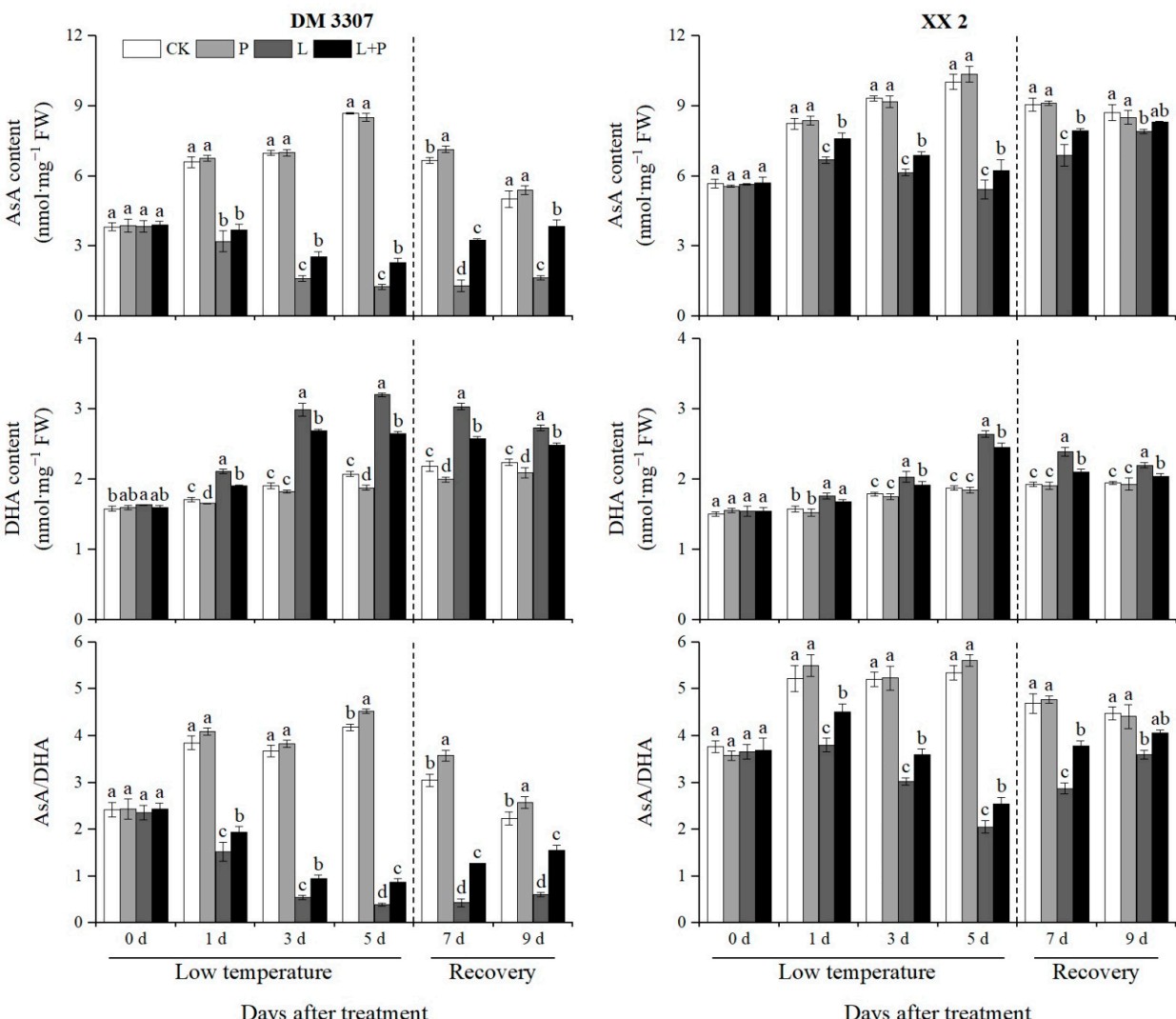

**Figure 4.** Effect of exogenous proline on the contents of AsA, DHA and AsA/DHA ratio in embryos of maize under the different treatment conditions. Notes: Data are expressed as mean ± standard deviation. Different letters within the same column indicate significant differences at the 5% level. CK: control group; P: proline treatment; L: low-temperature stress treatment; L + P: low-temperature stress + proline combined treatment. Different small letters represent significant differences between treatments ($p < 0.05$).

The contents of GSH and GSSH in DM 3307 and XX 2 embryos increased gradually with the extension of low-temperature stress time and reached the peak on the fifth day of treatment. On the first, third and fifth day of low-temperature stress, compared with the normal temperature control treatment, the GSH content of DM 3307 embryos treated with low temperature increased by 22.2%, 7.5% and 9.4%, respectively, and the GSSH content increased by 84.73%, 456.15% and 644.63%, respectively. The GSH content of XX 2 embryos increased by 37.67%, 55.85% and 43.84%, respectively, and the GSSH content increased by 61.66%, 217.10% and 426.52%, respectively. On the fifth day of low-temperature stress, compared with the treatment of low-temperature stress alone, the GSH content and GSH/GSSH of DM 3307 embryos soaked with proline increased by 42.3% and 120.1%, respectively, and the GSSH content decreased by 35.2%. The GSH content and GSH/GSSH of XX 2 embryos increased by 24.9% and 66.2%, respectively, and the GSSH content decreased by 25.0%. At the same time, after the release of low-temperature stress, the contents of GSH and GSSH of the two maize varieties decreased, while GSH/GSSH increased. On the ninth day of treatment, compared with low-temperature treatment, the GSH content of DM 3307 and XX

2 increased by 37.6% and 15.47%, respectively, and the GSSH content decreased by 17.7% and 15.4%, respectively under proline treatment (Figure 5).

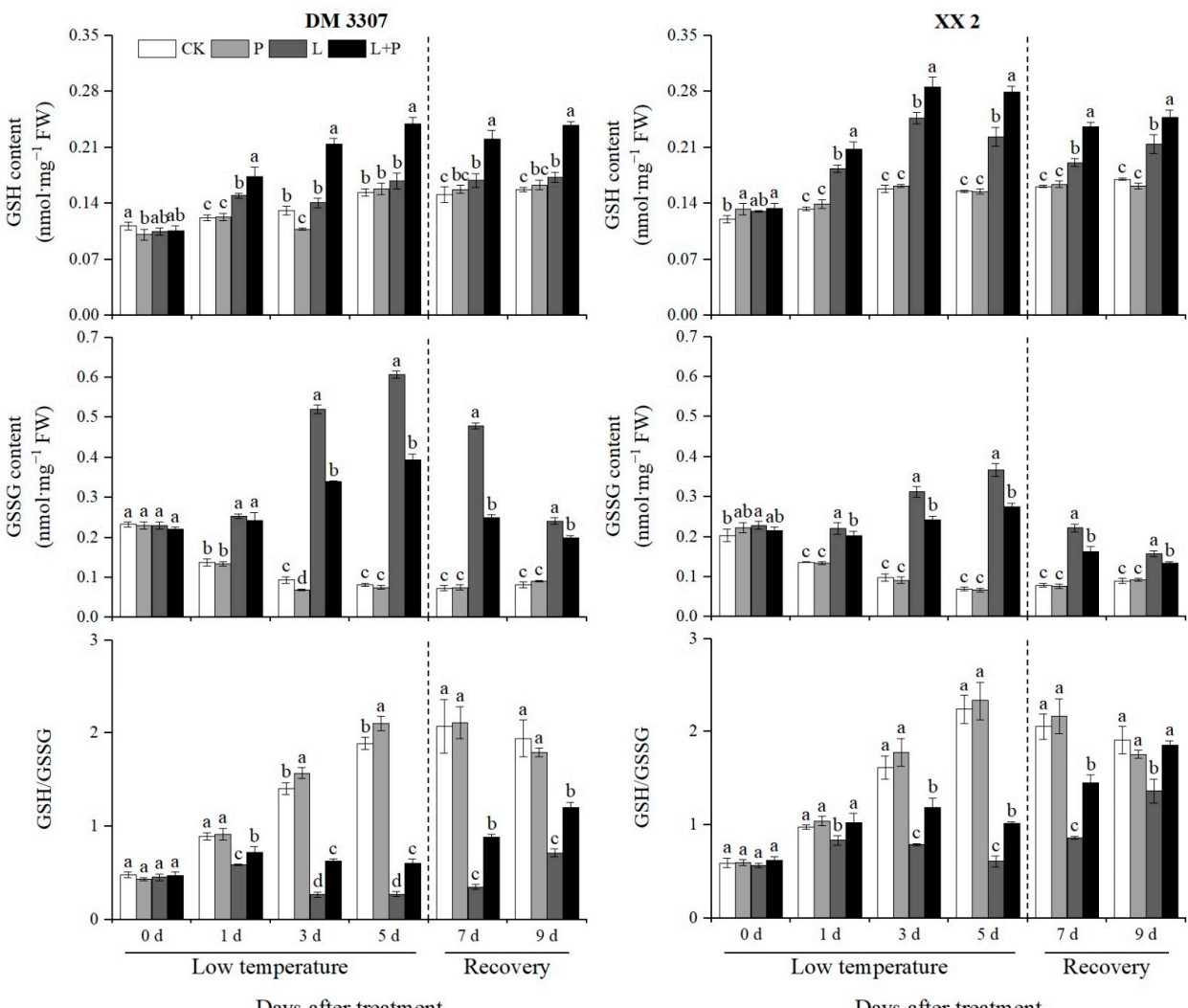

**Figure 5.** Effect of exogenous proline on the contents of GSH, GSSG and GSH/GSSG ratio in embryos of maize under the different treatment conditions. Notes: Data are expressed as mean ± standard deviation. Different letters within the same column indicate significant differences at the 5% level. CK: control group; P: proline treatment; L: low-temperature stress treatment; L + P: low-temperature stress + proline combined treatment. Different small letters represent significant differences between treatments ($p < 0.05$).

With the extension of low-temperature stress time, the activities of MDHAR and GR in the embryos of the two maize varieties increased gradually, the activity of APX increased first and then decreased, while the activity of DHAR decreased, reaching the extreme value on the fifth day of treatment. On the fifth day of treatment, compared with the normal temperature control treatment, low-temperature stress treatment significantly induced the increase of APX and GR activities of seed embryos of two maize varieties but inhibited the activities of MDHAR and DHAR. Among them, the decrease of DM 3307 was greater than that of XX 2. Under low-temperature stress, proline soaking treatment further promoted the activities of APX, MDHAR, DHAR and GR in the embryo of two maize varieties, and this promoting effect was more significant in DM 3307 variety. On the fifth day of low-temperature stress, compared with the treatment of simple low-temperature stress, the activities of APX, MDHAR, DHAR and GR in 3307 embryos of Damin increased by 15.94%,

86.33%, 36.14% and 41.87%, respectively. The activities of APX, MDHAR, DHAR and GR in XX 2 embryos increased by 10.03%, 36.76%, 47.07% and 15.21%, respectively. At the same time, after the low-temperature stress was relieved, the activities of APX, MDHAR, DHAR and GR in the embryos of the two maize varieties remained at a high level, and the promoting effect of seed soaking with proline was still obvious (Figure 6).

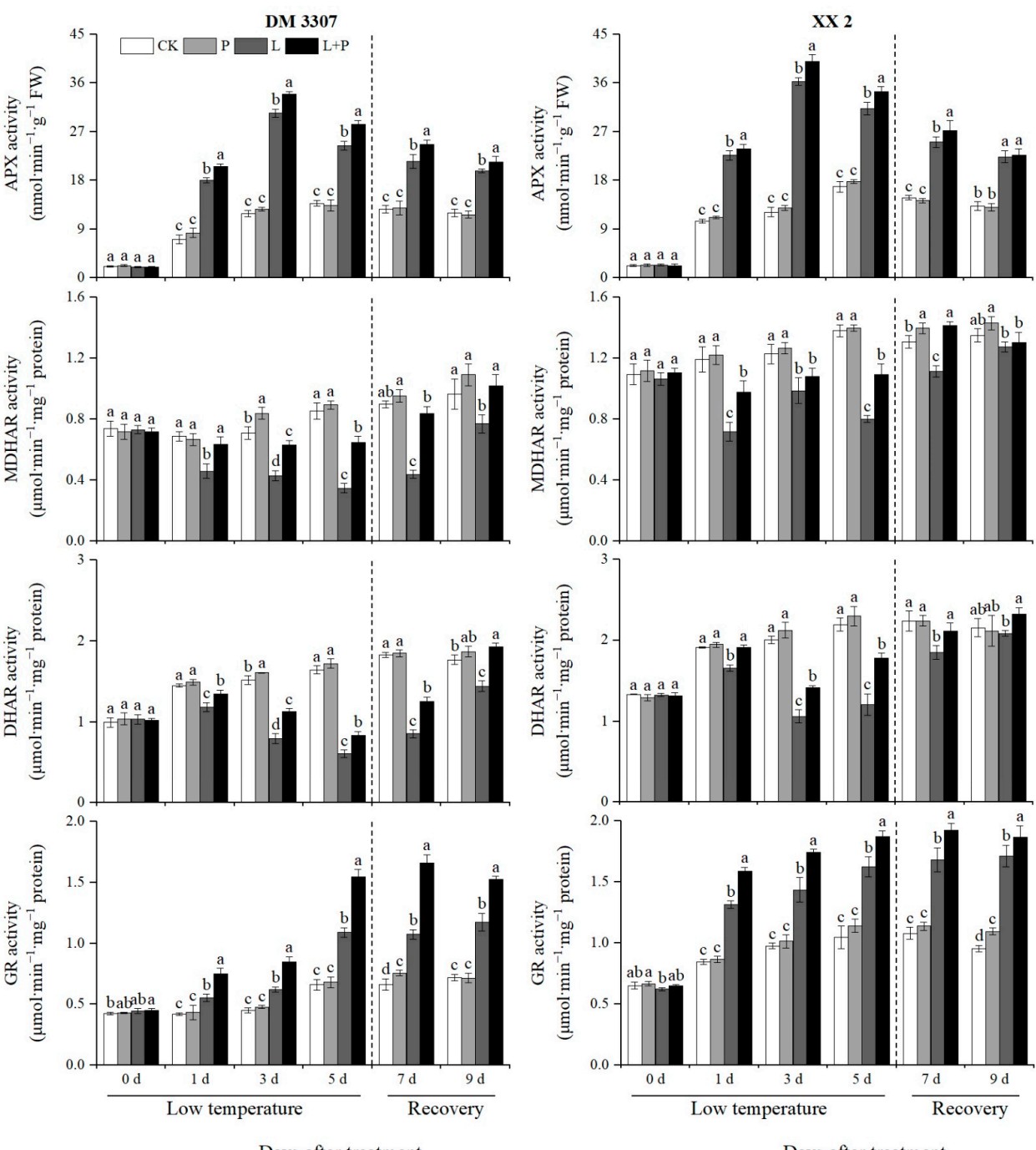

**Figure 6.** Effect of exogenous proline on the activity of APX, MDHAR, DHAR and GR in embryos of maize under the different treatment conditions. Notes: Data are expressed as mean ± standard deviation. Different letters within the same column indicate significant differences at the 5% level. CK: control group; P: proline treatment; L: low-temperature stress treatment; L + P: low-temperature stress + proline combined treatment. Different small letters represent significant differences between treatments ($p < 0.05$).

### 3.5. Soluble Protein and Soluble Sugar

With the extension of low-temperature stress time, the contents of soluble protein and soluble sugar in DM 3307 and XX 2 maize embryos increased first and then decreased, and reached the peak on the third day of treatment. At room temperature, seed soaking with proline could increase the content of soluble protein and soluble sugar in the embryo of two maize varieties, but it did not reach a significant level between treatments. On the third day of low-temperature stress, compared with the normal temperature control, the contents of soluble protein and soluble sugar in DM 3307 embryos increased by 20.20% and 18.85%, respectively, and the contents of soluble protein and soluble sugar in XX 2 embryos increased by 23.78% and 19.39%, respectively. Under low-temperature stress, seed soaking with proline can further improve the content of soluble protein and soluble sugar in maize embryos. On the third day of treatment, the soluble protein content of DM 3307 and XX 2 embryos treated with low-temperature stress combined with proline soaking increased by 17.23% and 14.17%, respectively, and the soluble sugar content increased by 16.02% and 8.96%, respectively. The results showed that exogenous proline treatment could induce the accumulation of osmoregulation substances under low-temperature stress, and the low-temperature sensitive variety DM 3307 was more significant. At the same time, the soluble protein and soluble sugar contents of the two maize varieties decreased by varying degrees after releasing the low temperature of the maize embryo. On the fourth day of low-temperature release, except for the soluble protein of DM 3307, the soluble protein and soluble sugar contents of proline seed soaking treatment were higher than those of low-temperature treatment, but the difference was not significant (Figure 7).

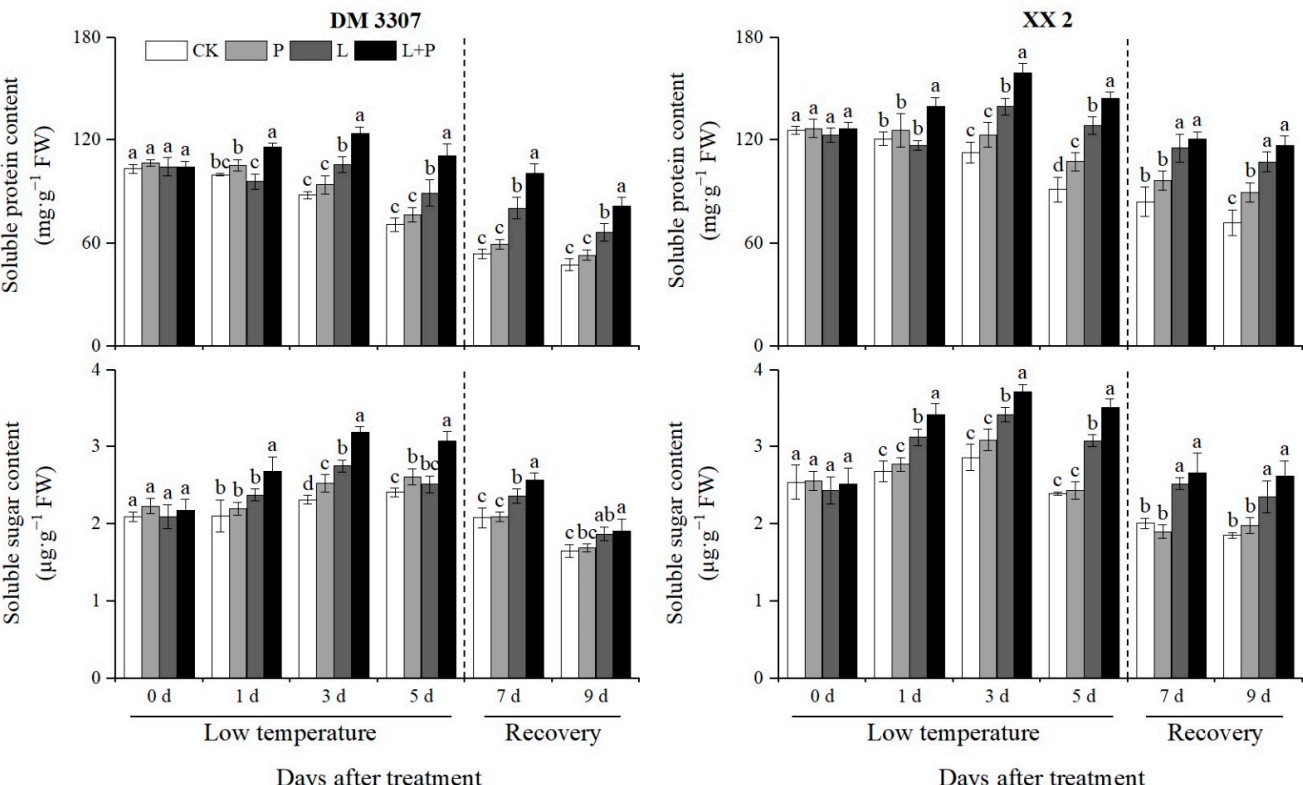

**Figure 7.** Effect of exogenous proline on the contents of soluble protein and soluble sugar in embryos of maize under the different treatment conditions. Notes: Data are expressed as mean ± standard deviation. Different letters within the same column indicate significant differences at the 5% level. CK: control group; P: proline treatment; L: low-temperature stress treatment; L + P: low-temperature stress + proline combined treatment. Different small letters represent significant differences between treatments ($p < 0.05$).

### 3.6. Metabonomic Analysis

3.6.1. Qualitative and Quantitative Analysis of Metabolites

The original mass spectrometry data of UPLC-MS/MS were processed by Analyst 1.6.3 software, and the metabolites of the sample were analyzed qualitatively and quantitatively by mass spectrometry based on the local metabolic database. The multi-peak map of MRM detection of multimodal maps of maize embryos is shown in Figure 8. By comparing the mass spectrometry data with the database and qualitative and quantitative analysis, 589 metabolites were detected in this study, including rich secondary metabolites in addition to primary metabolites, including amino acids, sugars and alcohols, organic acids, lipids, nucleotides and their derivatives, phenols, flavonoids and alkaloids. Qualitative metabolites in maize seed embryos were shown in Supplementary Table S1. Metabolites screened and Q1/Q3 transitions were shown in Supplementary Table S2.

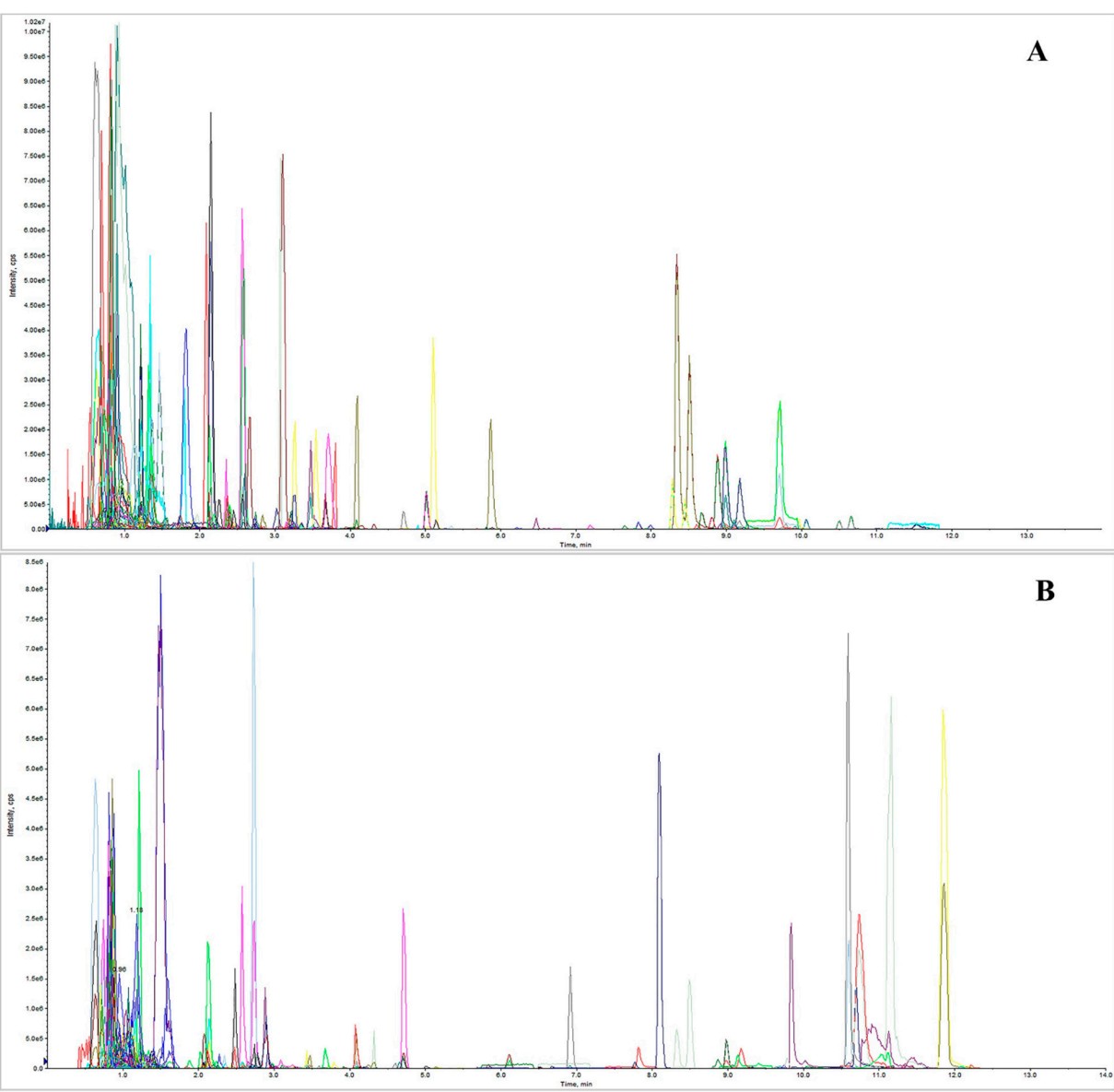

**Figure 8.** MRM detection of multimodal maps of maize seed embryos in the positive (**A**) and negative (**B**) ion mode. Notes: The abscissa is the retention time (RT) of metabolite detection, and the ordinate is the ion current intensity of ion detection (the intensity unit is CPS, count per second). Each mass spectrum peak with different colors represents a metabolite detected.

### 3.6.2. Overall Metabolic Observation of Samples

Unsupervised principal component analysis (PCA) was performed on all samples and the results were as follows. QC samples are made by mixing control, chilling and chilling + proline. It can be seen from the figure that QC samples are grouped into one class, which shows that they are robust and reliable rather than false positive. The difference between the treatments in the analysis chart is obvious, and there is a trend of "small settlement" among the repetitions. The interpretation rates of the first principal component and the second principal component to the data set are higher, which are 42.28% and 19.53%, respectively. The interpretation rate of the third principal component of the data set is not high, only 7.33% (Figure 9).

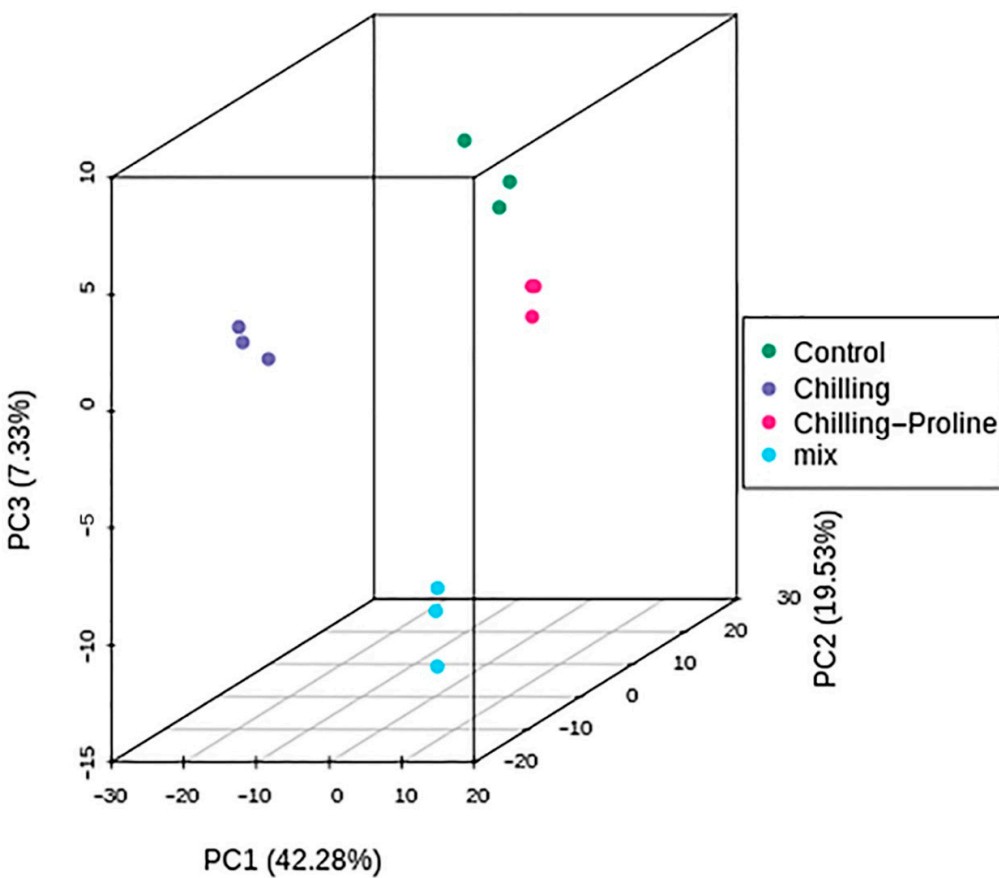

**Figure 9.** PCA score map of all samples.

### 3.6.3. Cluster Analysis of Metabolites

It can be seen from the relative content and cluster analysis of metabolites in the heat map that there are great differences among groups, especially in the relative content of metabolites between low-temperature stress treatment and normal temperature control treatment. Many metabolites with high relative content (Red Square) in the normal temperature control group are down-regulated under low-temperature stress, and many metabolites with low expression (Green Square) in normal temperature control group are up-regulated under low-temperature stress. Although there were differences in the relative contents of many metabolites between the low temperature + proline treatment group and the normal temperature control group, the exogenous proline treatment restored the expression of some metabolites affected by low-temperature stress. Metabolites are generally divided into four categories. The relative content of the first category of metabolites is lower in the low-temperature group. The second metabolite was highly expressed in the normal temperature control group. The relative content of the third metabolite was higher in the low-temperature group and the low temperature + proline treatment group.

This kind of metabolite was induced by low temperatures. The expression of the fourth metabolite was higher in the low temperature + proline treatment group (Figure 10).

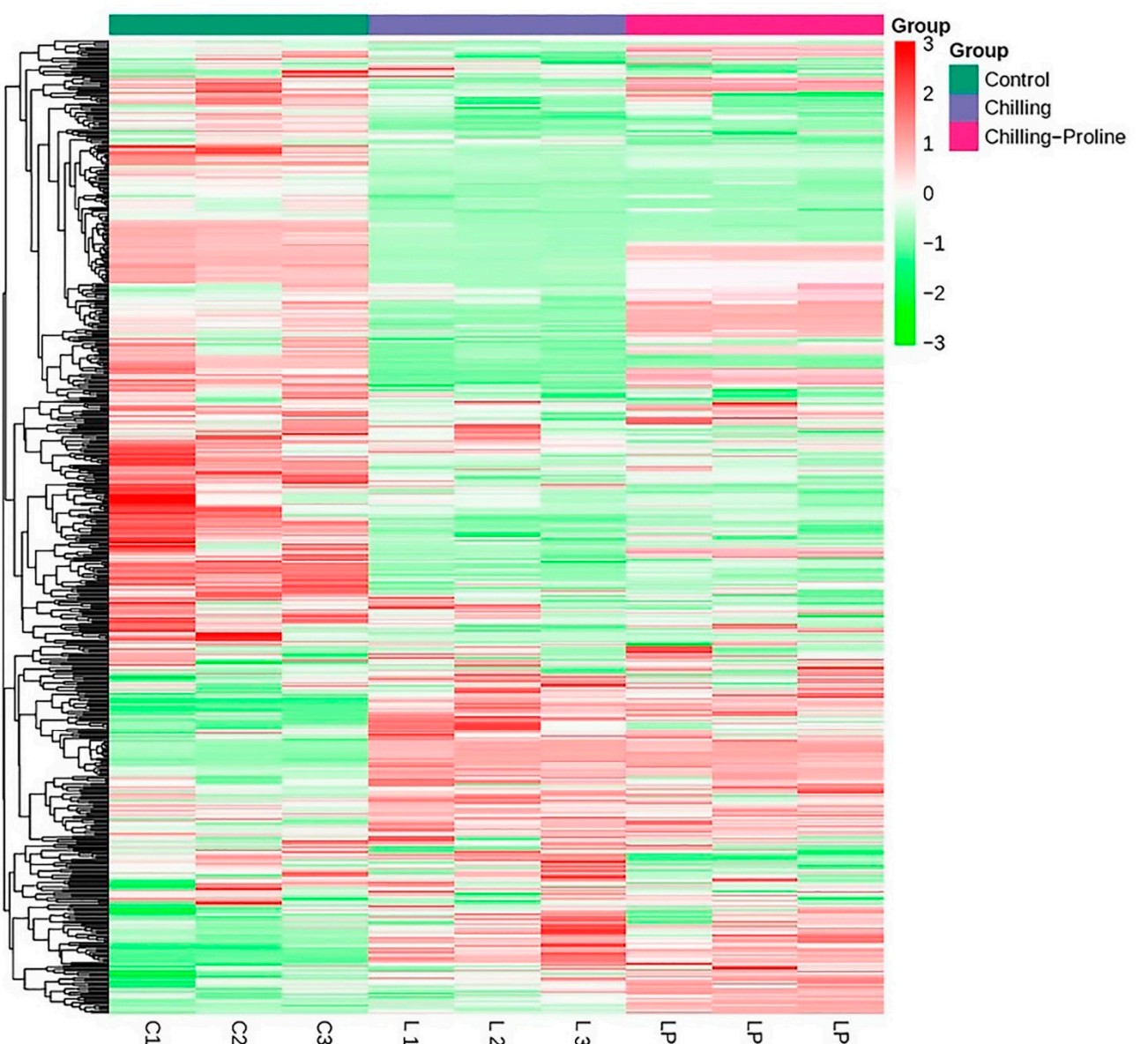

**Figure 10.** Heatmap and dendrogram of maize seed embryos under exogenous proline and low-temperature stress treatments.

### 3.6.4. Screening of Differential Metabolites

OPLS-DA model validation and univariate analysis were performed on the qualitative metabolite data in each treated sample. The metabolites with VIP $\geq$ 1 and $Log_2FC$ (fold change) $\geq$ 1 were selected as differential metabolites. The results showed that there were 262 different metabolites, including 32 organic acids, 28 amino acids, 20 nucleotides and their derivatives, 26 sugars and alcohols, 46 lipids, 51 alkaloids, 44 phenols and 15 other metabolites. The changing trend of the total content of several main differential metabolites is shown in Figure 11. From the overall trend, the total amount of various metabolites under low-temperature treatment was significantly lower than that under normal temperature control, while free fatty acids and phenolic acids were on the contrary. Compared with low temperature, the application of exogenous proline alleviated the decrease in the total amount of various metabolites (Figure 11).

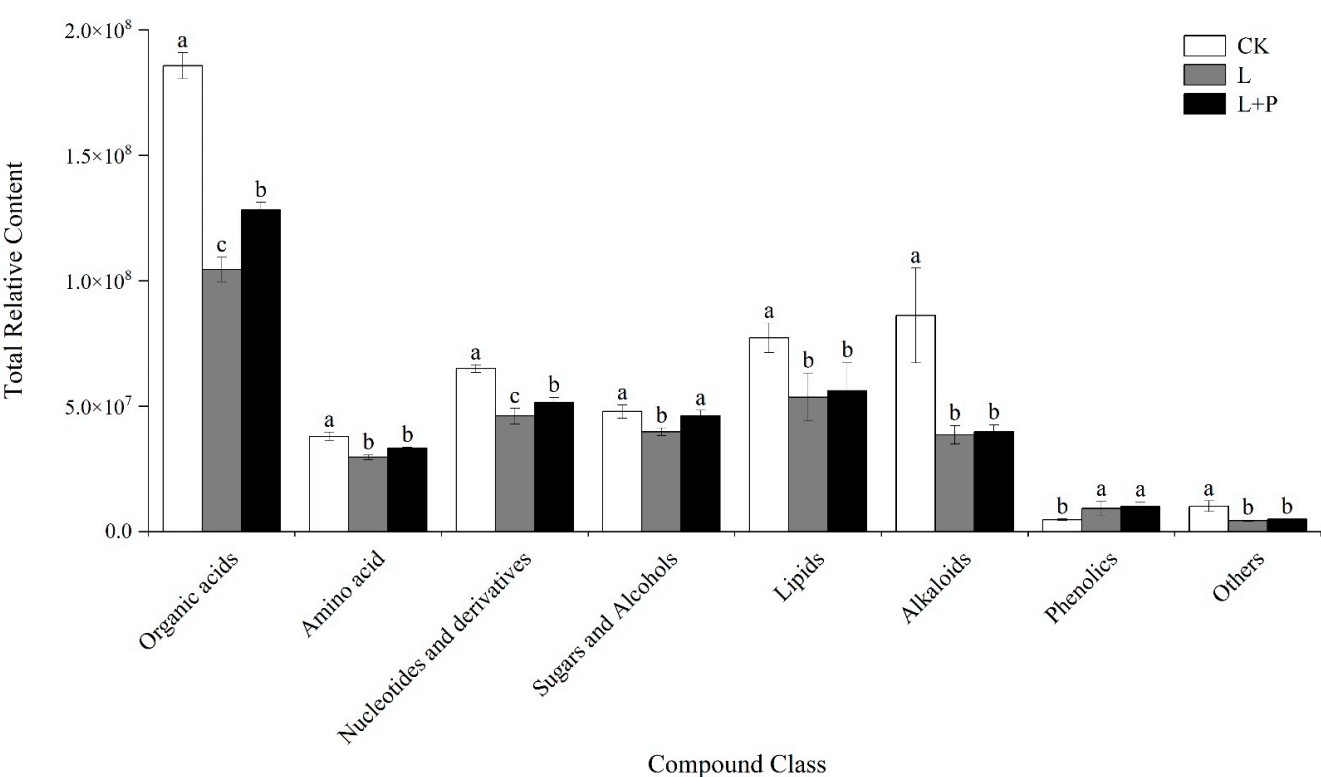

**Figure 11.** Changes in the total content of various major metabolites. Different small letters represent significant differences between treatments ($p < 0.05$).

3.6.5. Analysis of Differential Metabolites

Organic Acids

In this study, 32 organic acids were differentially expressed. Compared with the normal temperature group, there were 16 differentially expressed metabolites detected in maize embryos germinating at low temperature at 4 °C for 3 days (L) up-regulated with an increased range of 6.68–103.74%, such as azelaic acid. The other 16 differential organic acids were down-regulated, and the largest decreased organic acid was 2-pyridinecarboxylic acid, which decreased by 88.34%. The application of exogenous proline brought about extensive changes in organic acids of maize embryos metabolites different from those that occurred in response to low-temperature stress. Compared with the low-temperature stress group, the use of proline increased the relative contents of 19 organic acids whose relative content decreased under low-temperature stress, such as ami-nomalonic acid, α-ketoglutaric acid, azelaic acid, 2-picolinic acid, trans-citridic acid. According to the heatmap, low-temperature stress affects the expression of organic acids, and exogenous proline can alleviate the effect of low-temperature stress on the expression of these organic acids (Figure 12).

Amino Acids and Derivatives

The 28 differentially expressed amino acids and derivatives were Homomethionine, Proline, Asparagine, N-(3-indolylacetyl)-l-alanine, Glycine, Histidine, N-propionylglycine, Phenylacetyl-L-glutamine, N-glycyl-L-leucine, S-adenosylmethionine, theanine, Alanyl-L-phenylalanine, Glycyl-L-isoleucine, Glycyl-L-phenylalanine, Lysine-butanoic acid, N-acetyl-l-methionine, N-acetyl-l-tryptophan, Phenylalanine, N-acetyl-l-glutamic acid, N6-acetyl-l-lysine, N-acetyl-l-arginine, N-acetyl-L-leucine, Histamine, Ornithine, N-acetyl-l-aspartic acid, S-adenosyl-l-methionine, N-acetyl-L-threonine and S-ribosyl-L-homocysteine. Proline changed the most significantly, increasing by 5.11 times. The application of exogenous proline alleviated the effect of low-temperature stress on amino acid metabolism. Compared with the low-temperature stress group, the use of proline increased the relative content of 19 amino acids in embryos under low-temperature stress, among which glycine and

proline increased by 43.57% and 44.53%, respectively. The application of exogenous proline brought about extensive changes in amino acids and derivatives of maize embryo metabolites different from those that occurred in response to low-temperature stress. Compared with the low-temperature stress group, the use of proline increased the relative contents of 19 amino acids and derivatives, including Phenylacetyl-L-glutamine, Histamine, N-acetyl-L-tryptophan, Alanyl-L-phenylalanine, N-acetyl-L-methionine, Glycine, N-propionylglycine, N-(3-indolylacetyl)-L-alanine, Lysine-butanoic acid, N-acetyl-L-glutamic acid, N-acetyl-L-aspartic acid, N-acetyl-L-leucine, Homomethionine, Theanine, Ornithine, Phenylalanine, N6-acetyl-L-lysine, Proline and Histidine. It can be seen from the heat map that low-temperature stress affects the expression of amino acids, and exogenous proline can significantly alleviate the effect of low-temperature stress on the expression of these metabolites (Figure 13).

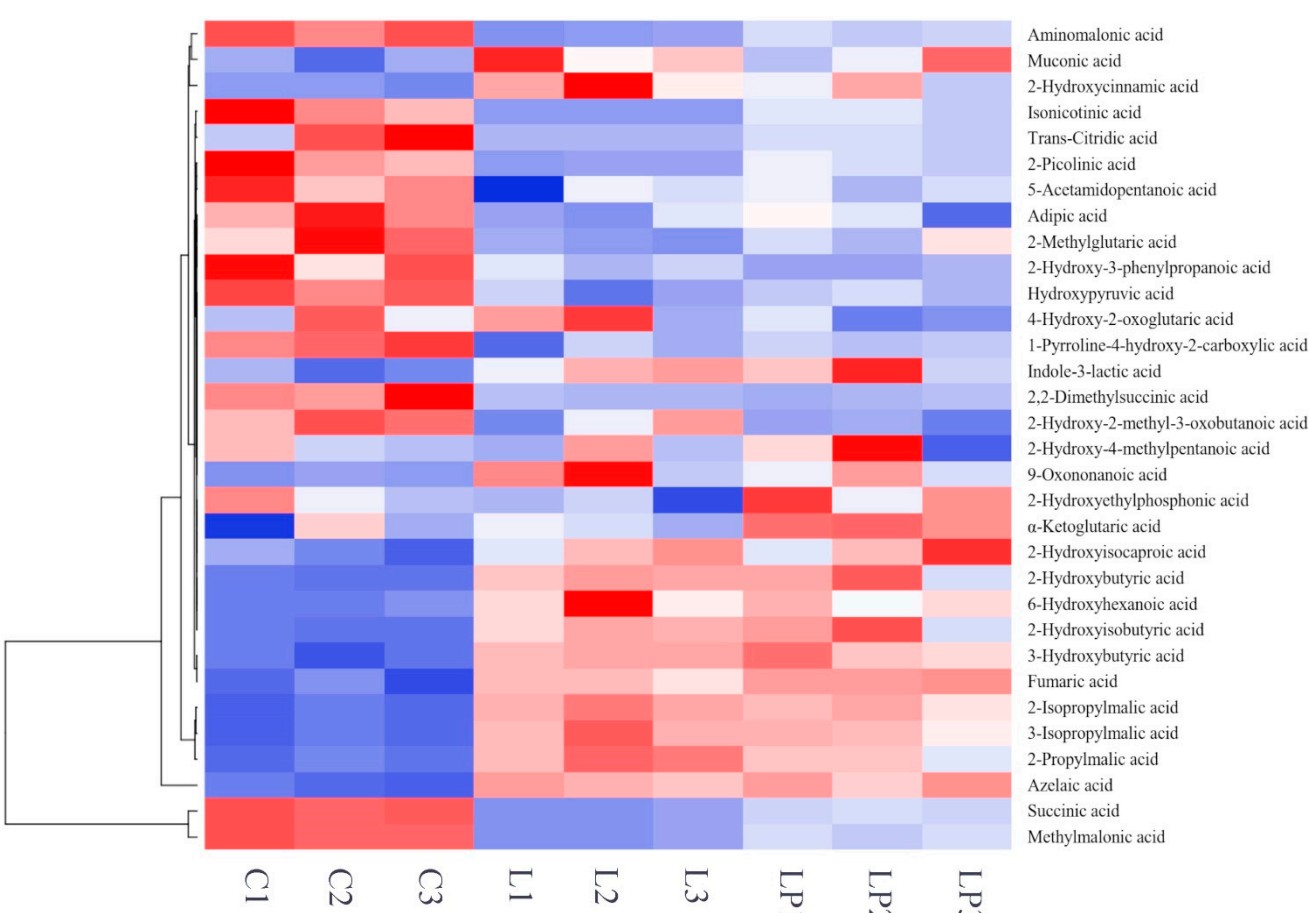

**Figure 12.** Heatmap and dendrogram of organic acid metabolites in embryos of maize under exogenous proline and low-temperature stress treatments.

Nucleotides and Derivatives

In this study, a total of 20 nucleotides and derivatives were differentially expressed, including 2′-deoxycytidine, 7-methylxanthine, 2-aminopurine, 5-methyluridine, uridine 5′-diphosphate, cytidylic acid, ribosyladenosine, guanosine 5′-monophosphate, isopentenyladenine-7-n-glucoside, 2′-deoxyadenosine, cytarabine, 2-deoxyribose-1-phosphate, 2-(dimethylamino)guanosine, adenosine 5′-monophosphate, xanthine, guanine, nicotinic acid adenine dinucleotide, 2′-deoxyinosine-5′-monophosphate, 6-methylmercaptopurine

and 5′-deoxy-5′-(methylthio)adenosine. Compared with the normal temperature group, there were 14 differentially expressed nucleotides and derivatives detected in maize embryos germinating at low temperature at 4 °C for 3 days (L) up-regulated and 6 down-regulated, with a change multiple of 0.35–3.35. The relative content of guanine changed most significantly, increasing by 234.74%. Compared with the low-temperature stress group, the use of proline increased the relative contents of 12 differentially expressed nucleotides and derivatives, xanthine changed most significantly, increasing by 47.35%. It can be seen from the heatmap that low-temperature stress affects the expression of nucleotides and their derivatives, and exogenous proline can alleviate the effect of low stress on the expression of these metabolites (Figure 14).

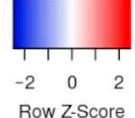

**Figure 13.** Heatmap and dendrogram of amino acids metabolites in embryos of maize under exogenous proline and low-temperature stress treatments.

Lipid Metabolites

In this study, a total of 46 lipids were differentially expressed. This includes 13 kinds of free fatty acids, 5 kinds of glycerol esters, 27 kinds of lysophospholipids and 1 kind of terpenoids. Compared with the normal temperature group, there were 9 differentially expressed lipids detected in maize embryos germinating at low temperature at 4 °C for 3

days (L) up-regulated and 42 down-regulated, with a change multiple of 0.09–3.48. The application of exogenous proline increased the relative content of 27 lipids under low-temperature stress. It contains 23 kinds of lysophosphatidylcholine, 3 kinds of glyceryl esters and one kind of terpenoid, among which LysoPC 20:2 was the most significantly up-regulated, increasing by 142.07 %. Exogenous application of proline significantly alleviated the effect of low-temperature stress on lipid metabolism (Figure 15).

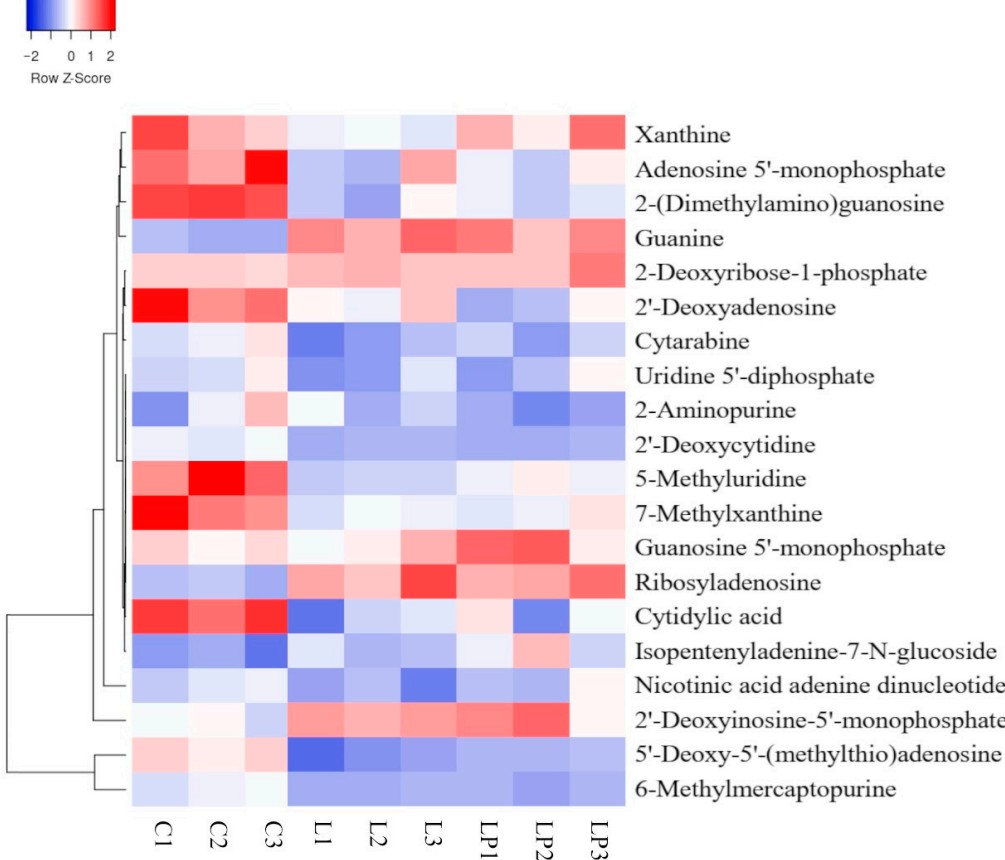

**Figure 14.** Heatmap and dendrogram of nucleotides and derivatives metabolites in embryos of maize under exogenous proline and low-temperature stress treatments.

Alkaloids Metabolites

The most differentially expressed metabolites were alkaloids, with 51 species. Compared with the normal temperature group, the relative contents of 44 alkaloids decreased and 7 alkaloids increased under low-temperature stress, with a multiple of 0–5.11. The largest increase was dicaffeoyl spermidine. Exogenous application of proline alleviated the effect of low-temperature stress on alkaloid metabolism. Compared with low-temperature stress treatment, the use of proline increased the relative content of 19 alkaloids and decreased 32 alkaloids under low-temperature stress. Among them, coconut amidopropyl betaine increased by 99.88%. Compared with low-temperature stress treatment, this shows that low-temperature stress affects the synthesis of alkaloids, and exogenous proline can significantly alleviate the effect of low-temperature stress on the synthesis of these metabolites (Figure 16).

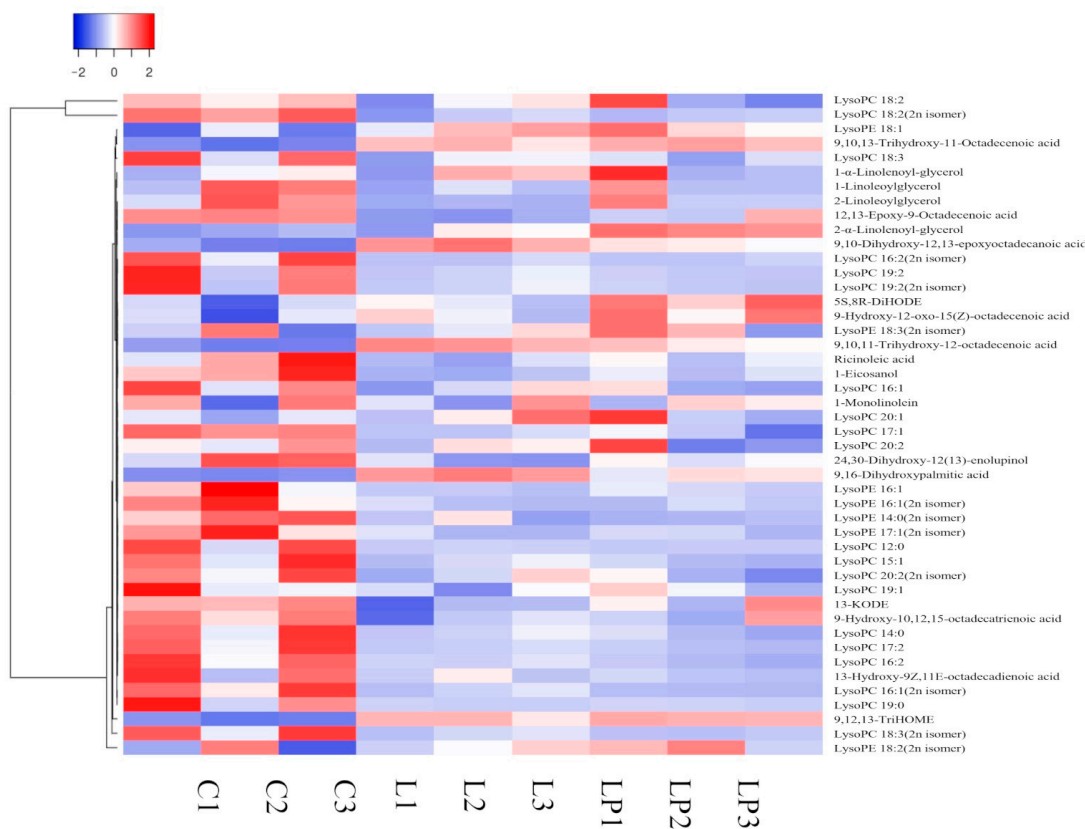

**Figure 15.** Heatmap and dendrogram of lipids metabolites in embryos of maize under exogenous proline and low-temperature stress treatments.

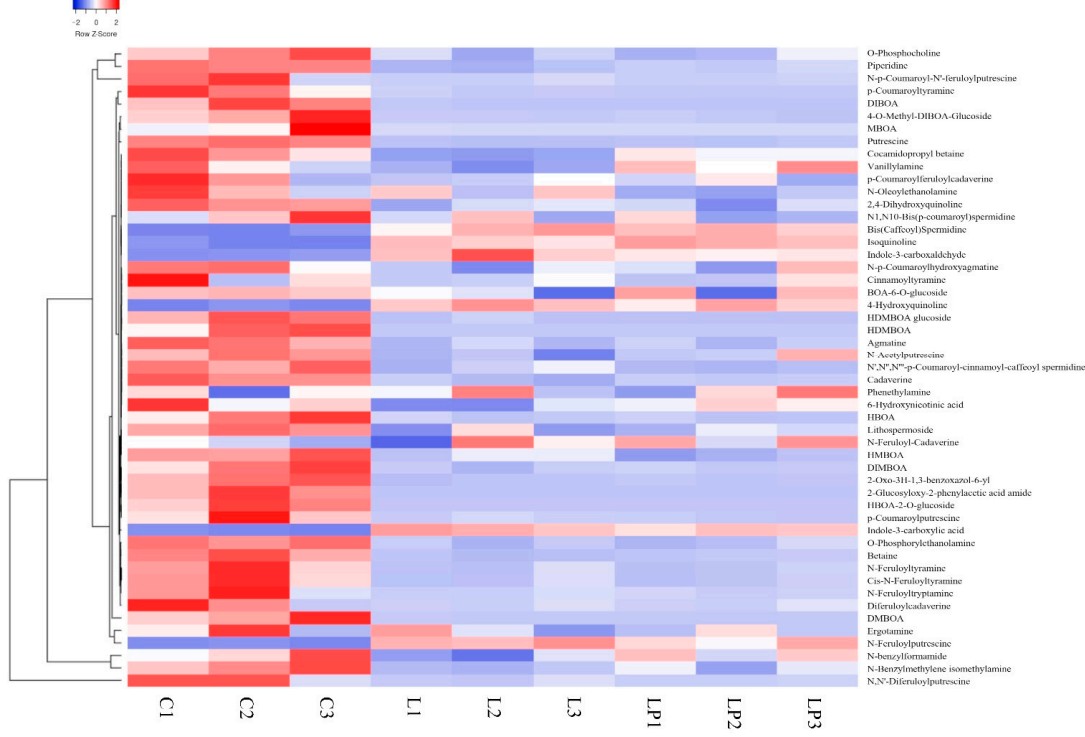

**Figure 16.** Heatmap and dendrogram of alkaloids metabolites in embryos of maize under exogenous proline and low-temperature stress treatments.

Phenols Metabolites

In this study, 44 phenolic metabolites were differentially expressed, including 13 flavonoids, 28 phenolic acids and 3 coumarins. Compared with the normal temperature group, the relative contents of 12 flavonoids decreased and 1 increased under low-temperature stress, with a multiple of 0.03–1.26. Exogenous proline significantly alleviated the effect of low-temperature stress on flavonoid metabolism. Compared with the low-temperature stress group, the use of proline increased the relative content of 11 flavonoids under low-temperature stress. The relative contents of myricetin-3-O-glucoside, naringin-7-O-glucoside and catechin were up-regulated most significantly, with a multiple of 2.23–4.12. Among the 31 differentially expressed phenolic acids and coumarins, 19 were up-regulated and 12 were down-regulated under low-temperature stress, with a multiple of 0.14–6.40. 3-hydroxycinnamic acid changed the most significantly, increasing by 540.46%. The application of exogenous proline changed the effects of low-temperature stress on the relative contents of phenolic acid and coumarin. Compared with the low-temperature stress group, the use of proline increased the relative contents of 23 phenolic acids and coumarins under low-temperature stress and decreased the contents of 8 metabolites, among which chlorogenic acid increased the most significantly, increased by 231.09%. The relative content of 6-O-feruloyl-d-glucose, isoferulic acid and other metabolites decreased. This shows that low-temperature stress inhibits the expression of flavonoids, some phenolic acids and coumarins. Phenolic metabolites such as 3-hydroxycinnamic acid are enriched under low-temperature stress in response to low-temperature stress. Exogenous proline can significantly alleviate the effect of low-temperature stress on the expression of these metabolites (Figure 17).

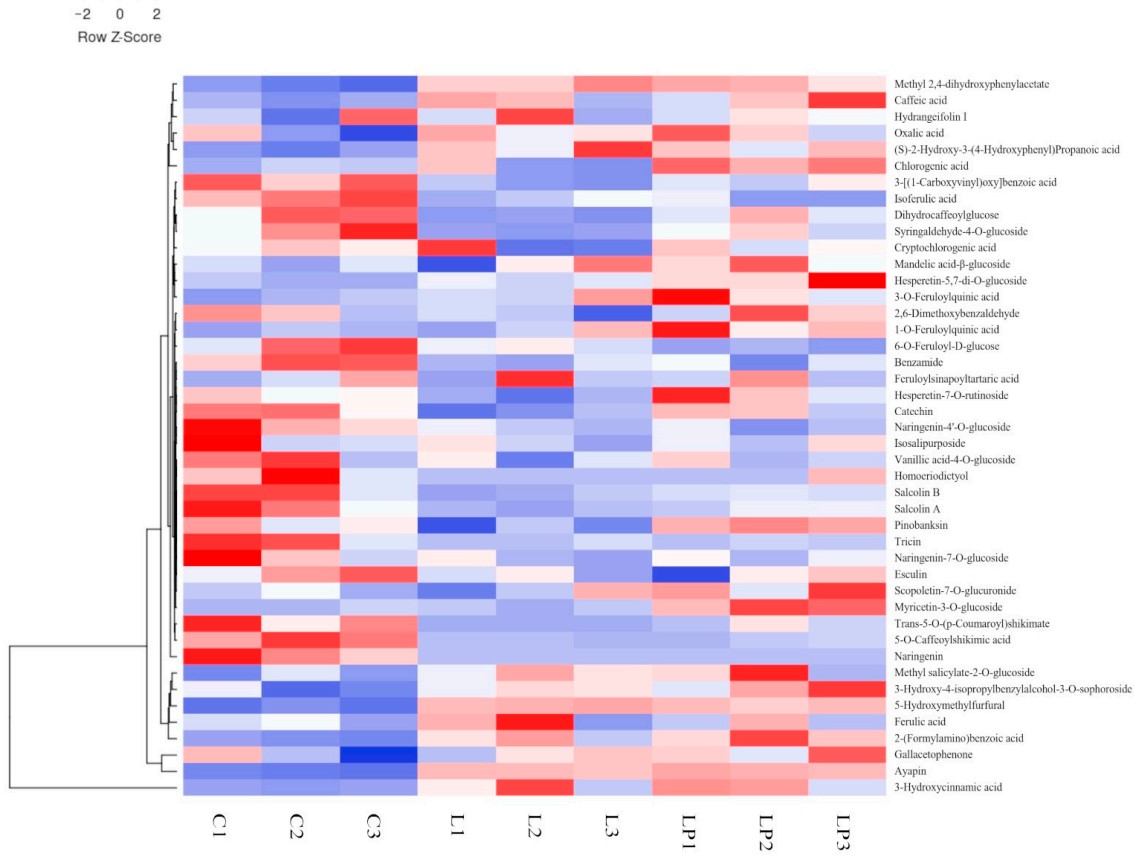

**Figure 17.** Heatmap and dendrogram of phenols metabolites in embryos of maize under exogenous proline and low-temperature stress treatments.

Vitamins and Other Metabolites

In this study, the other 15 metabolites differentially expressed contained 8 vitamins. Compared with the normal temperature group, the relative contents of ascorbic acid, niacin and other metabolites decreased and pyridoxine-5-phosphate increased under low-temperature stress. The application of exogenous proline alleviated the effect of low-temperature stress on vitamin metabolites but had no significant effect on other metabolites. Compared with the low-temperature stress group, the use of proline increased the relative content of all seven vitamin metabolites under low-temperature stress, among which niacin and ascorbic acid changed most significantly, increasing by 230.90% and 113.26%, respectively. This shows that low-temperature stress affects the synthesis of vitamins, and exogenous proline can significantly alleviate the effect of low-temperature stress on the synthesis of these metabolites (Figure 18).

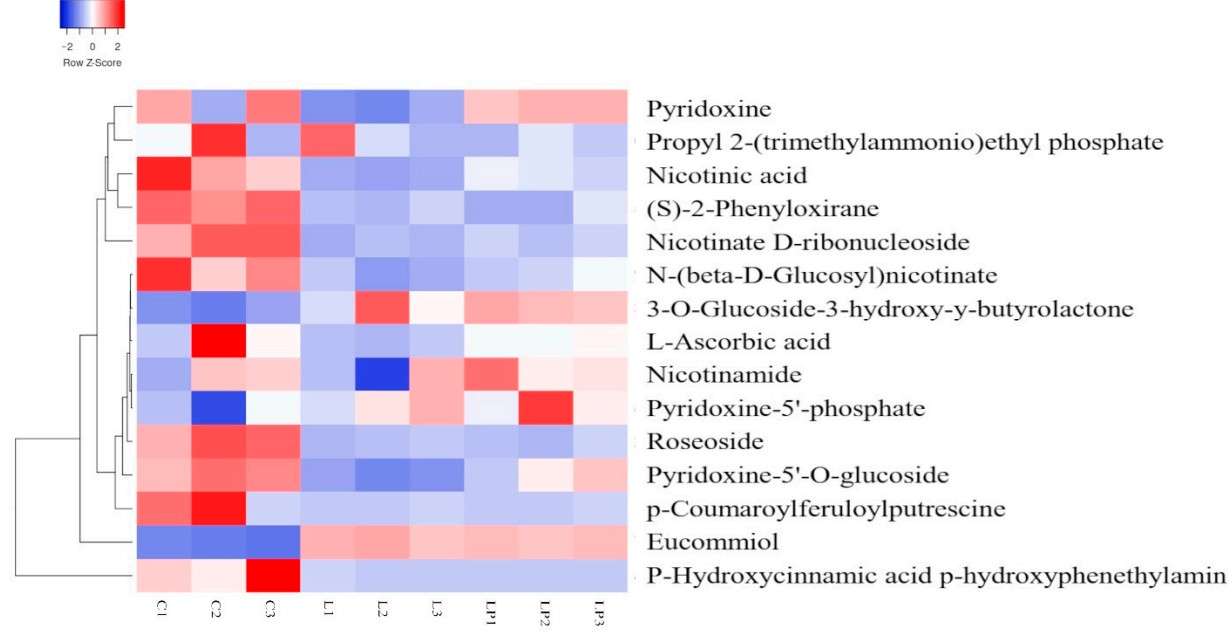

**Figure 18.** Heatmap and dendrogram of vitamins and other metabolites in embryos of maize under exogenous proline and low-temperature stress treatments.

3.6.6. KEGG Analysis of Differential Metabolites

Pathway enrichment analysis of differential metabolites can better study the mechanism of pathway change. The main signal pathways involved in differential metabolites were obtained by KEGG enrichment analysis, and a total of 68 metabolic pathways were obtained. The five pathways with the most significant changes are: starch and sucrose metabolism, arginine and proline metabolism, biosynthesis of secondary metabolites, flavonoid biosynthesis and the pentose phosphate pathway (Table 1).

**Table 1.** List of pathway analysis in the embryos of maize under low-temperature stress.

| Number | Pathway Name | Pathway ID | Number | Pathway Name | Pathway ID |
|--------|-------------|------------|--------|-------------|------------|
| 1 | Glycolysis/Gluconeogenesis | ko00010 | 35 | Starch and sucrose metabolism | ko00500 |
| 2 | Citrate cycle (TCA cycle) | ko00020 | 36 | Amino sugar and nucleotide sugar metabolism | ko00520 |
| 3 | Pentose phosphate pathway | ko00030 | 37 | Polyketide sugar unit biosynthesis | ko00523 |

**Table 1.** *Cont.*

| Number | Pathway Name | Pathway ID | Number | Pathway Name | Pathway ID |
|---|---|---|---|---|---|
| 4 | Pentose and glucuronate interconversions | ko00040 | 38 | Glycerolipid metabolism | ko00561 |
| 5 | Fructose and mannose metabolism | ko00051 | 39 | Inositol phosphate metabolism | ko00562 |
| 6 | Galactose metabolism | ko00052 | 40 | Glycerophospholipid metabolism | ko00564 |
| 7 | Ascorbate and aldarate metabolism | ko00053 | 41 | Linoleic acid metabolism | ko00591 |
| 8 | Synthesis and degradation of ketone bodies | ko00072 | 42 | α-Linolenic acid metabolism | ko00592 |
| 9 | Ubiquinone and other terpenoid-quinone biosynthesis | ko00130 | 43 | Sphingolipid metabolism | ko00600 |
| 10 | Oxidative phosphorylation | ko00190 | 44 | Pyruvate metabolism | ko00620 |
| 11 | Arginine biosynthesis | ko00220 | 45 | Glyoxylate and dicarboxylate metabolism | ko00630 |
| 12 | Purine metabolism | ko00230 | 46 | Propanoate metabolism | ko00640 |
| 13 | Caffeine metabolism | ko00232 | 47 | Butanoate metabolism | ko00650 |
| 14 | Pyrimidine metabolism | ko00240 | 48 | C5-Branched dibasic acid metabolism | ko00660 |
| 15 | Alanine, aspartate and glutamate metabolism | ko00250 | 49 | Carbon fixation in photosynthetic organisms | ko00710 |
| 16 | Glycine, serine and threonine metabolism | ko00260 | 50 | Vitamin B6 metabolism | ko00750 |
| 17 | Monobactam biosynthesis | ko00261 | 51 | Nicotinate and nicotinamide metabolism | ko00760 |
| 18 | Cysteine and methionine metabolism | ko00270 | 52 | Pantothenate and CoA biosynthesis | ko00770 |
| 19 | Valine, leucine and isoleucine degradation | ko00280 | 53 | Zeatin biosynthesis | ko00908 |
| 20 | Lysine biosynthesis | ko00300 | 54 | Sulfur metabolism | ko00920 |
| 21 | Lysine degradation | ko00310 | 55 | Phenylpropanoid biosynthesis | ko00940 |
| 22 | Arginine and proline metabolism | ko00330 | 56 | Flavonoid biosynthesis | ko00941 |
| 23 | Histidine metabolism | ko00340 | 57 | Isoflavonoid biosynthesis | ko00943 |
| 24 | Tyrosine metabolism | ko00350 | 58 | Stilbenoid, diarylheptanoid and gingerol biosynthesis | ko00945 |
| 25 | Phenylalanine metabolism | ko00360 | 59 | Tropane, piperidine and pyridine alkaloid biosynthesis | ko00960 |
| 26 | Tryptophan metabolism | ko00380 | 60 | Glucosinolate biosynthesis | ko00966 |
| 27 | Phenylalanine, tyrosine and tryptophan biosynthesis | ko00400 | 61 | Aminoacyl-tRNA biosynthesis | ko00970 |

**Table 1.** *Cont.*

| Number | Pathway Name | Pathway ID | Number | Pathway Name | Pathway ID |
|---|---|---|---|---|---|
| 28 | Benzoxazinoid biosynthesis | ko00402 | 62 | Biosynthesis of various secondary metabolites—part 2 | ko00998 |
| 29 | β-Alanine metabolism | ko00410 | 63 | Metabolic pathways | ko01100 |
| 30 | Taurine and hypotaurine metabolism | ko00430 | 64 | Biosynthesis of secondary metabolites | ko01110 |
| 31 | Phosphonate and phosphinate metabolism | ko00440 | 65 | Carbon metabolism | ko01200 |
| 32 | Cyano amino acid metabolism | ko00460 | 66 | 2-Oxocarboxylic acid metabolism | ko01210 |
| 33 | D-Arginine and D-ornithine metabolism | ko00472 | 67 | Biosynthesis of amino acids | ko01230 |
| 34 | Glutathione metabolism | ko00480 | 68 | ABC transporters | ko02010 |

## 4. Discussion

In the process of plant growth, there is a dynamic balance between ROS and the antioxidant system in plant cells. low-temperature stress will increase the permeability of plant cell membrane, a large amount of ROS accumulation, protein damage, membrane lipid peroxidation, nucleotide degradation, metabolic disorder, and finally cause the death of some cells and tissues [69]. In this study, low-temperature stress induced ROS production, as well as $O_2^{\bullet-}$ generation rate and $H_2O_2$ content. Excessive reactive oxygen species accelerated the rate of membrane peroxidation, resulting in serious electrolyte leakage and MDA production. With the extension of stress time, the degree of cell oxidative damage deepened. The excessive accumulation of reactive oxygen species showed that the embryo was subjected to low-temperature stress. The oxidative damage of DM 3307 was more serious than that of XX 2. Exogenous proline alleviated the oxidative damage in maize embryos caused by low-temperature stress and had a better effect on the cell membrane damage of DM 3307. After the stress was relieved, proline could accelerate the balance between ROS and its scavenging system in maize embryos of the two varieties. Wilmsen et al. [70] also found that proline can alleviate the oxidative damage caused by drought stress in citrus. Proline in plants can effectively scavenge reactive oxygen species through its own circulation and form stable free radical adducts. $H_2O_2$, as a signal molecule, initiates the phosphorylation cascade involving MKK4/5 and MPK3/6 to activate ROS scavenging enzymes, so as to scavenge reactive oxygen species [71]. In the results of this study, proline may play the role of active oxygen scavenger and membrane stabilizer to reduce ROS accumulation. In addition, exogenous proline also increased the enzyme activity and antioxidant content of the antioxidant system, which may be the performance of proline scavenging ROS by activating the antioxidant enzyme system through $H_2O_2$.

With the extension of low-temperature stress time, ROS accumulation and outbreak, plants will be regulated by various enzyme and non-enzyme antioxidant systems in vivo, which is of great significance to protect the integrity of the membrane. SOD is the most important enzyme for cell defense against ROS. SOD can directly convert $O_2^{\bullet-}$ to $H_2O_2$, which is catalytically decomposed by POD and CAT to produce $H_2O$ [71]. In this study, with the increase of $O_2^{\bullet-}$ generation rate and $H_2O_2$ content, the antioxidant enzyme activity increased first and then decreased under low-temperature stress, which was generally higher than that of the normal temperature control. After the stress was relieved, the enzyme activities tended to be stable. The SOD of the two varieties increased most significantly on the first day, which may be due to $O_2^{\bullet-}$ increase and SOD is activated first as the first line of defense of the antioxidant system. With the extension of low-temperature stress time, on the fifth day, the activities of various antioxidant enzymes began to decrease,

and the activity of CAT decreased to a level lower than normal temperature. The increase of CAT activity in DM 3307 was not significant, contributing less to ROS clearance under low-temperature stress. However, the continuous increase of $H_2O_2$ and MDA content indicates that the enhancement of antioxidant enzyme activity induced by low-temperature stress is not enough to remove excess ROS and repair oxidative damage. The application of exogenous proline further enhanced the activities of three antioxidant enzymes and reduced the accumulation of ROS. This result is consistent with the findings of Chen et al. [72]. Exogenous proline up-regulated the gene expression encoding SOD (*Cu/ZnSOD*, *MnSOD*), APX (*CytAPX*) and CAT (*CatC*) in salt-stressed rice [42]. The results of this study confirmed the view that proline as an exogenous substance can regulate the enzyme activity of the antioxidant system.

AsA-GSH is an important antioxidant circulatory system in plants, including antioxidant enzymes such as APX, MDHAR, DHAR and GR, as well as non-enzymatic antioxidants AsA and GSH, which are mainly responsible for removing excess $H_2O_2$ in plant cells [73]. Under abiotic stress, the coordinated operation of various components in the AsA-GSH cycle can eliminate the reactive oxygen species accumulated in plants and protect cell tissues from the damage of reactive oxygen free radicals. It is an important way to maintain the normal division and growth of cells [74]. At the same time, the activities of APX and GR are usually regarded as the directional enzymes of plant tolerance to stress conditions, and DHAR and MDHAR are the guarantees of the AsA regeneration cycle [75]. The ratio of AsA/DHA and GSH/GSSG is the signal that plants regulate the dynamic change of ROS. Therefore, the efficiency of the AsA-GSH cycle can be reflected by the ratio of AsA/DHA and GSH/GSSG [76]. In our study, low-temperature stress significantly decreased the content of ASA, the activities of MDHAR and DHAR and the ratio of AsA/DHA and GSH/GSSG, increased the content of DHA, GSH and GSSG and the activity of GR, which was proportional to the stress time. APX activity increased first and then decreased with the extension of stress time. After the stress was relieved, the AsA-GSH cycle of XX 2 recovered rapidly, and the recovery of DM 3307 was slow. This is due to the increase of APX activity before and during low temperatures, which mobilized AsA to participate in the clearance of $H_2O_2$. However, due to the decrease of reductase MDHAR and DHAR activities, DHA increased and GSH/GSSG ratio decreased, resulting in the obstruction of the AsA cycle. With the deepening of stress, the cells were damaged in the later stage, APX activity decreased and the efficiency of the AsA-GSH cycle further decreased. Under low-temperature stress, the contents of AsA and GSH increased, the activities of GR, MDHAR and DHAR increased significantly, the ratio of AsA/DHA and GSH/GSSG increased, and the contents of DHA and GSSH decreased, among which the activities of MDHAR and DHAR increased most significantly. The mitigation effect of DM 3307 is more significant. This is consistent with the findings of previous studies on chickpeas under temperature stress [77]. These findings clearly show that proline can promote the efficiency of the AsA-GSH cycle under low-temperature stress by up-regulating the activity of antioxidant enzymes, improving the antioxidant level, effectively removing excess ROS, reducing the membrane damage and membrane lipid peroxidation of embryo cells under low-temperature stress, and accelerating the recovery of balance of reactive oxygen species metabolism after relieving stress.

Metabolites are downstream products of gene expression and participate in plant metabolism. Their types and quantities are significantly related to plant phenotypes [78]. Metabonomics is the simultaneous qualitative and quantitative analysis of all small molecule metabolites of organisms or cells in a specific physiological period. It is an important part of systems biology [79]. In recent years, with the gradual development and improvement of detection technology and corresponding databases, metabonomics has been widely used to screen and mine important metabolites and metabolic pathways in the process of plant growth and resistance to stress [80–82]. So far, the defense process of metabonomics under stress has become the focus of extensive research, mainly involving some model plants and cash crops, and there is less research on crops [83]. Plant metabolites are generally divided

into primary metabolites and secondary metabolites, which are closely related to plant stress resistance [84]. By analyzing the changes of plant metabolites under stress conditions, we can find the functions of some unknown genes or proteins, which provides more basis for revealing the mechanism of the plant stress response. Through the metabolome analysis of Arabidopsis under low-temperature stress by GC-MS technology, it was found that the contents of aspartic acid, ornithine and arginine in plants increased in response to low temperature [85]. Extensively targeted metabonomics was used to study peanut metabolites under low-temperature stress. It was found that long-term low temperature would reduce most metabolites, while plant sphingosine, proline, putrescine and aspartic acid would increase.

Through the metabolomic analysis of maize embryos, we found that the content of a large number of metabolites decreased under low-temperature stress, but some metabolites also increased. Sugar is an important carbon source and energy source for plants. It not only provides energy for life activities but also the intermediate product of sugar metabolism is the main carbon skeleton. Under low-temperature stress, important osmotic stress substances of sugar crops will accumulate in large quantities, which can protect proteins and cell membranes [86]. In this study, sorbitol, planteose, melezitose, trehalose-6-phosphate and sugar alcohols were significantly up-regulated after medium and low-temperature stress. Hexoses accumulation is very important to improve the low-temperature resistance of plants and has been confirmed in Arabidopsis, wheat and other plants [85,87]. In this study, planteose reached 3.58 times that of the control group. The glycolytic intermediates decreased significantly. The RNA component ribose also decreased significantly. The above results show that low temperature significantly inhibits sugar metabolism and reduces the supply of materials and energy required for embryo development. However, during low temperature, the embryo adapts to the low-temperature environment by enriching planteose, melezitose, d-(-)-threose and other sugars and sugar alcohols. Under the treatment of exogenous proline, the relative contents of the above differential sugars and sugar alcohols were generally increased, especially for ribose, d-(-)-threose, erythritose 4-phosphate and arabinose. Arabinose is an aldose involved in the composition of cell wall components. Erythritose 4-phosphate and ribose are the intermediate products of pentose phosphate pathway. The increase of erythritose 4-phosphate and ribose indicates that PPP is activated and accelerated. PPP provides material and energy for seed embryo cells under low-temperature stress to maintain the basic metabolic needs of seed embryos. Hexoses are the most important carbohydrate in plant metabolism. The above results show that exogenous proline can increase the content of osmoregulatory substances by increasing the enrichment of sugars and sugar alcohols in response to low-temperature stress, such as ribose, planteose, erythritose 4-phosphate and arabinose. It can activate the sugar metabolism efficiency of seed embryos under low-temperature stress as a whole, especially improve the PPP operation efficiency and provide more reducing power for seed embryo metabolism, so as to enhance the tolerance of embryo to low temperature.

The content of organic acids in plant cells under adverse stress can reflect the strength of plant growth and metabolic activity [84]. In this study, low-temperature stress reduced the total relative content of differential organic acids. Stress will reduce succinic acid and enrich fumaric acid [88]. However, there is no research on the relationship between fumaric acid and plant stress resistance. α-ketoglutarate is not only an important intermediate product of the TCA cycle, coordinating the carbon and nitrogen metabolism system, but also closely related to plant energy metabolism and the synthesis of vitamins, proteins and amino acids [89]. α-ketoglutarate provides a carbon skeleton for ammonium assimilation, enters the tricarboxylic acid cycle and generates proline [90]. α-ketoglutarate can improve plant stress resistance and regulate nitrogen metabolism [91]. α-ketoglutarate is related to low temperature promoting proline metabolism. Trans-aconitic acid can be catalyzed by aconitic acid isomerase to transform with CIS aconitic acid, which is a metabolic branch of the TCA cycle. Previous studies have found that trans-aconitic acid is very effective in killing pests, and CIS aconitic acid is an important intermediate product in

the TCA cycle [92]. The application of exogenous proline increased the content of total organic acids, and the content of 19 differential organic acids increased. TCA cycle intermediates α-ketoglutarate, succinic acid and fumaric acid were enriched, indicating that exogenous proline can promote the recovery of the TCA cycle. The above results showed that exogenous proline could increase the content of most organic acids in embryos under low-temperature stress, especially during the recovery of the TCA cycle.

Previous studies have shown that low temperature will reduce the content of amino acids in seeds. This study obtained consistent results. low-temperature stress reduced the total content of 28 differential amino acids and their derivatives in seed embryos. Compared with the control group, there were only three kinds of amino acids enriched under low-temperature stress, namely proline, asparagine and N-(3-indoleacetyl)-L-alanine. The application of exogenous proline increased the total content of 19 differential amino acids and their derivatives. The increase of ornithine shows that exogenous proline can increase the content of endogenous proline by increasing the ornithine pathway. At the same time, the ornithine cycle can detoxify ammonia and store ammonia. The ornithine cycle is also linked to the TCA pathway through fumaric acid, which is released to supplement the TCA cycle [93]. Glycine is an effective nitrogen source, which can be transformed into pyruvate and enter the TCA cycle as an amino acid skeleton [94]. The above results showed that exogenous proline increased the content of amino acids in seed embryos under low-temperature stress, mainly through the enrichment of proline and glycine, promoted proline metabolism, reduced the ammonia toxicity of seed embryos under low-temperature stress, and improved the low-temperature resistance of plants.

Nucleotides are precursors of nucleic acid biosynthesis. Some nucleotides are raw materials for synthesizing active intermediates, and some nucleotides are important metabolic regulators. In this study, four nucleotides and their derivatives were enriched under low-temperature stress, including guanine, riboadenosine, 2-deoxyinosine-5-monophosphate and isopentenyl adenosine-7-n-glucoside. The application of exogenous proline increased the relative content of 15 nucleotides and their derivatives in maize embryos under low-temperature stress. Xanthine with the highest change amplitude increased by 1.47 times. This showed that exogenous proline promoted purine metabolism under low-temperature stress. Intermediates of purine metabolism, such as acylureas, are low consumption compounds with a high N/C ratio, which play an important role in the survival of plants under stress [95].

Lipids can be divided into three categories according to their components: simple lipids (fatty acids, fats and waxes), complex lipids (phospholipids, sphingomyelins and glycosphingolipids) and unsaponifiable lipids (steroids and terpenes). Fatty acids are the main components of biofilms. Biofilm plays an important role in maintaining the stability of the intracellular environment, energy conversion and information transmission. Previous studies have shown that the effect of low-temperature stress on plants first occurs in the cell membrane system. Low temperature will cause membrane phase transition, weaken fluidity, shrink and increase membrane permeability. In severe cases, the cell membrane will rupture, cause metabolic disorder and even cause the death of cells and tissues [96]. Membrane lipids mainly include phospholipids, glycolipids, thiolipids and sterols. In this study, low-temperature stress significantly reduced the total content of lipids in seed embryos. Among the 51 differential lipids, 42 showed a downward trend. The most obvious type of decline was lysophosphatide (LP), with the highest decline of 92.51%. Lysophospholipid is the product of phospholipid hydrolysis. It is not only a phospholipid metabolite but also a signal molecule to regulate cell life activities including cell division, elongation and differentiation [97]. Under stress conditions, lysophospholipids can also activate the second signal molecule of $Ca^{2+}$, which plays a positive role in improving the stress tolerance of animals and plants [98]. In this study, the content of 26 of 27 differential lysophosphatides decreased due to low temperature. This indicates that low-temperature stress slows down the metabolic efficiency of phospholipids. The increase of unsaturated fatty acids at low temperatures can improve the fluidity of the cell membrane. Previous

studies have confirmed that plant tolerance is related to membrane lipid components, and the unsaturation of fatty acids is positively related to low-temperature tolerance [99,100]. Fatty acid dehydrogenase (FAD) is the key enzyme to produce unsaturated fatty acids. The expression of the fad gene was up-regulated in cotton under low-temperature stress. The increase of unsaturated fatty acid content is a positive response of plants to low-temperature stress [101]. Under low-temperature stress, seven unsaturated fatty acids such as 9 (S), 12 (S), 13 (S)-trihydroxy-10 (E)-octadecenoic acid, 9,10,13-trihydroxy-11-octadecenoic acid and 9,10,11-trihydroxy-12-octadecenoic acid were enriched in maize embryo, with an increase multiple of 1.21–3.48. Exogenous proline treatment increased the content of 27 kinds of lipids under low-temperature stress, including 26 kinds of unsaturated fatty acids. 2- α- glyceryl linolenate increased by 112.01%. In this study, only one terpene was differentially expressed. Under low-temperature stress, 24,30-dihydroxy-12 (13) ene lupin alcohol decreased significantly, and exogenous proline increased its content, but not significantly. The above results showed that the content of unsaturated fatty acids in seed embryos increased under low-temperature stress. Exogenous proline effectively protected the integrity of the cell membrane by reducing the degree of osmotic stress, reducing membrane lipid peroxidation damage, and accelerating the metabolic efficiency of phospholipids by increasing the precursors of cell membrane synthesis. The content of unsaturated fatty acids increased, so as to improve the fluidity of the cell membrane. The increase of phospholipid hydrolysates after exogenous proline treatment may be related to the fact that proline improves the overall metabolic efficiency of seed embryos.

Alkaloid is a secondary metabolite synthesized from amino acids. Alkaloids have different effects on plant growth and development. Many alkaloids are effective components of medicinal plants. Alkaloids also play an important role in resisting stress. Previous studies have found that lead stress can increase the accumulation of alkaloids in Catharanthus roseus [102]. In this study, five alkaloids were enriched after low-temperature stress, among which dicaffeoyl spermidine changed the most, increasing by 5.14 times. The application of exogenous proline increased the relative content of 27 alkaloids in maize embryos under low-temperature stress. Coconut amidopropyl betaine, vanillin amine, 6-hydroxynicotinic acid and BOA-6-O-glucoside changed significantly. At present, the research on these metabolites mainly focuses on industry and chemistry, and there are few reports on plants. However, these alkaloids can be used as nitrogen pools to improve sufficient substrates for seed germination. Phenolic compounds are secondary metabolites in plants, and their content and structural composition are closely related to environmental conditions [103]. Wilmsen et al. [70] showed that phenols in citrus fruits can scavenge DPPH free radicals. In this study, low-temperature stress significantly increased the total content of phenols in seed embryos. Twenty phenols such as 3-hydroxycinnamic acid and Eupatorium lactone were significantly up-regulated. Among them, 3-hydroxycinnamic acid increased by 540.46%. Exogenous proline treatment increased the total content of phenols under low-temperature stress, but it was not significant. The contents of 35 phenols including chlorogenic acid and myricetin-3-O-glucoside were up-regulated, and the higher increases were Eupatorium lactone and chlorogenic acid. This is consistent with the previous research results on fennel. Spraying 20 mmol $L^{-1}$ proline solution on the leaf can significantly increase the content of polyphenols in fennel leaves under drought stress [103]. However, the regulation mechanism of these phenols induced by low temperature and the regulation mechanism of proline need to be further explored. The relative content of ascorbic acid decreased significantly under low-temperature stress by 71.12%. The application of exogenous proline increased the content of ascorbic acid under low-temperature stress by 113.26%. This result is consistent with previous studies [104]. It shows that exogenous proline can alleviate the consumption of ascorbic acid under low-temperature stress, activate the ascorbic acid cycle, remove excess ROS, improve the antioxidant level of seed embryo under low temperature, and reduce the oxidative damage of seed embryo cells under low-temperature stress. These significantly different metabolites are mainly involved in the following metabolic pathways: starch and sucrose metabolism, arginine

and proline metabolism, biosynthesis of secondary metabolites, flavonoid biosynthesis and pentose phosphate pathway. The specific metabolic mechanism and biological significance of these potential biomarkers and metabolic pathways still have exploration value and need to be further studied.

## 5. Conclusions

Low-temperature stress enhanced the activities of SOD, POD, APX and GR in maize embryos, and proline soaking treatment further enhanced the activities of the above enzymes under stress. Proline seed soaking treatment inhibited the decrease of CAT, MDHAR and DHAR activities caused by low-temperature stress, increased the content of antioxidant substances, effectively reduced the accumulation of ROS, reduced the degree of membrane lipid peroxidation, maintained the integrity of cell membrane structure and reduced the oxidative damage of low-temperature stress to seed embryo. Through the statistical and cluster analysis of different metabolites among the treatment groups, it was found that exogenous proline mainly improved the low-temperature resistance of germinated maize embryos by enhancing the metabolism of starch and sucrose, arginine and proline, the biosynthesis of secondary metabolites, flavonoid biosynthesis and pentose phosphate pathway.

**Supplementary Materials:** The following supporting information can be downloaded at: https://www.mdpi.com/article/10.3390/pr10071388/s1, Table S1: Qualitative metabolites in maize seed embryos; Table S2: Metabolites screened and Q1/Q3 transitions.

**Author Contributions:** Conceptualization, J.L. and W.G.; methodology, S.Z.; software, S.Z.; validation, J.L.; formal analysis, S.Z.; investigation, Y.Z.; resources, S.W.; data curation, S.Z. and Y.Z.; writing original draft preparation, S.Z.; writing—review and editing, J.L. and S.W.; visualization, J.L., S.W. and W.G.; supervision, J.L; project administration, S.W. and W.G; funding acquisition, J.L. and W.G. All authors have read and agreed to the published version of the manuscript.

**Funding:** This work was supported by the National Natural Science Foundation of China (32172112) and the Special Project of Collaborative Innovation and Extension System of Green Organic Agriculture in Heilongjiang Province (GOAS2022).

**Institutional Review Board Statement:** Not applicable.

**Informed Consent Statement:** Not applicable.

**Data Availability Statement:** The data presented in this study are available within the article.

**Conflicts of Interest:** The authors declare no conflict of interest.

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
