# Peer review of "Exogenous Proline Optimizes Osmotic Adjustment Substances and Active Oxygen Metabolism of Maize Embryo under Low-Temperature Stress and Metabolomic Analysis"

_processes, doi:10.3390/pr10071388_

Round 1

Reviewer 1 Report

The study on Exogenous Proline Optimizes Osmotic Adjustment Substances and Active Oxygen Metabolism of Maize Embryo under Low Temperature Stress and Metabolomic Analysis, is of great interest. The analysis done was extensive and shows the response pattern of maize embryo under the treatment conditions.   

1. The report of the materials and method should be reported in past tense

2. The Line numbering is supposed to start from the beginning, instead it started from section 3,6,2

2. The use of 0.1% HgCl2 solution to disinfect the seed for 10min, seem to be to much exposure the chemical which might reduce the seeds viability and affect the test result. about 2 to 3 min is a good duration to expose the seed.

3. Are you reffering to this as oxide ion O2•−? what is the dot (•) for?

4. In the charts, can you add the meaning of CK,P, L etc along side, to easily remember what they represent.

5. ''Effect of exogenous proline on the contents of AsA, DHA and AsA/DHA ratio in embryos of maize under low temperature stress.'' 6. The title of  Figure 4, as with other should not be written as ........'under low temperature stress' alone, since there are treatments including exposure to proline under normal temperature and a control. it is better to put it this way; ........... under the different treatment conditions.   7. Also, the figures showing the results, was labeled low temperature and recovery, but there are other treatments there that were not under low temperature, an appropriate label should be given

Author Response

Dear Reviewer 1,

Thank you for the helpful and valuable comments revising the manuscript. We have studied comments carefully and revised portions are marked in red in the manuscript.

Comment 1:The study on Exogenous Proline Optimizes Osmotic Adjustment Substances and Active Oxygen Metabolism of Maize Embryo under Low Temperature Stress and Metabolomic Analysis, is of great interest. The analysis done was extensive and shows the response pattern of maize embryo under the treatment conditions.   

Reply 1:Thank you very much for your recognition and appreciation of our research topics. Your encouragement is our greatest motivation. In the future, our research team will continue to work hard to explore the mechanism of Exogenous Proline enhancing the ability of Maize to resist low temperature stress from the perspective of physiology, ecology and molecular biology.

Comment 2:The report of the materials and method should be reported in past tense

Reply 2:Indeed, as you said, grammatical tenses need to be improved. We have made changes and checked and improved the sentence spelling, grammar and number marking of the manuscript. Best.

Comment 3:The Line numbering is supposed to start from the beginning, instead it started from section 3,6,2

Reply 3:We apologize for wrong line numbering in our manuscript. We have made changes to the manuscript. Thank you.

Comment 4:The use of 0.1% HgCl2 solution to disinfect the seed for 10min, seem to be to much exposure the chemical which might reduce the seeds viability and affect the test result. about 2 to 3 min is a good duration to expose the seed.

Reply 4:Thank you for pointing out this important issue. We have revised to make method succinct to the point. Best.

Comment 5:Are you reffering to this as oxide ion O2•? what is the dot () for?

Reply 5:We refer to the relevant references, O2 generation rate is a conventional expression. () means nothing, maybe it means “importance”.

Comment 6:In the charts, can you add the meaning of CK,P, L etc along side, to easily remember what they represent.

Reply 6:Thank you for your thoughtful suggestions. We have modified and improved the notes below the tables and figures in the manuscript so that readers can better understand our research content.

Comment 7:''Effect of exogenous proline on the contents of AsA, DHA and AsA/DHA ratio in embryos of maize under low temperature stress.'' The title of  Figure 4, as with other should not be written as ........'under low temperature stress' alone, since there are treatments including exposure to proline under normal temperature and a control. it is better to put it this way; ........... under the different treatment conditions. 

Reply 7:Based on your helpful and careful suggestions,we have revised and improved the sentences in the manuscript to present them in a clearer way, especially deleting some wordy expressions and repeated sentences.

Comment 8:Also, the figures showing the results, was labeled low temperature and recovery, but there are other treatments there that were not under low temperature, an appropriate label should be given

Reply 8:Thank you for your suggestions. This is an important issue. We have modified and improved the manuscript.

We are very grateful for you raising above valuable comments for improving manuscript. After receiving your helpful comments yesterday, our scientific research and academic team held an emergency video conference with Tencent conference software immediately. We have studied your valuable comments carefully and we hope our reply could meet with approval. If you have any other questions, please feel free to contact us.

Best regards.

Shiyu Zuo, Yuetao Zuo, Wanrong Gu, Shi Wei and Jing Li

10 June,2022

Reviewer 2 Report

The manuscript submitted by Zuo et al is devoted to investigation of the effect of proline in maize seed germination. The germination was studied for seeds kept under different conditions and different media.

Having considered the submission, I have following comments and questions:

0) Line numbers are absent, so , it may turn out to be tricky to find the text mentioned in my review.

1) Page 1, abstract, bottom line: I'm not sure that it is correct to classify the mentioned substances as alkaloids.

2) Page 3: The comma after H2O2 seems to be corrupted.

3) Page 4, 1O2: please use superscript and subscript.

4) Page 5, section 2.2: I think that it is better to say "distilled water" instead of "0 mmol*L-1".

5) Here and after throughout the text: Please correct the style of material representing. Here, I mean using "The above seed soaking solution was placed..." instead "Place the above seed soaking solution...".

6) Page 5, section 2.3.1: "1 mL supernatant was added with 0.9 mL..." - does it mean " was mixed with"?

7) Page 8, section 2.3.8.2: The caption is apparently incorrect. LC-MS/MS method was used but not GC/MS.

8) Here: "the injection volume is 4 μL。" - please use a correct dot symbol.

9) the end of the section 2.3.8.2: The description of the mass spectrometer parameters needs a revision as it contains some extra information (e.g. PPG concentrations used for calibrations) along with some contradictions (CAD gas is mentioned to be high and medium). In addition to this, I recommend to provide a supplementary table containing the parameters for MRM transitions of the metabolites. Moreover, it is not clear whether a single experiment with a switching polarity was used or two different experiments were made in POS and NEG mode. Please clarify it here.

10) Section 2.3.8.3: The whole section should be revised thoroughly. The authors state that they conducted not only qualitative analysis but quantification of the analytes as well. The latter is impossible without building hundreds of calibrations for each metabolite separately. I guess that the authors have performed peak integration and between-groups comparison of peak areas (so-called Fold change).

11) Page 10, Fig. 1: Here and further: what do the letter a,b,c above the bars mean? The legend should be given either within the diagrams or below.

12) Page 12, section 3.4: ASA and AsA is used for ascorbic acid through the text. Please make it uniform.

13) Page 13: Please use capitals here for GSSH and check whether it is typed correctly elsewhere in the text.

14) Page 17, Fig. 8: Actually, these are not TIC chromatograms but MRM chromatograms recorded in the experiments.

15) Page 19, section 3.6.5.1: I'm curious to learn how authors could separate the hexoses using a C18 chromatographic column. This is also in the support of my request to provide a table containing all MRM parameters and, maybe, retention times of the metabolites.

16) Page 21, section 3.6.5.3: Please correct the font size here to make it uniform.

Most of my comments are of typesetting except those on the description of the MS/MS analysis. I'd like to learn more details on the metabolite screening and separation of polar highly soluble metabolites on a C18 column. The latter can influence the results of metabolite profiling and, this metabolic pathways enrichment.

Author Response

Dear Reviewer 2,

Thank you for helpful and valuable comments revising manuscript. We have studied comments carefully and revised portion are marked in red in the manuscript.

Comment 1:Line numbers are absent, so , it may turn out to be tricky to find the text mentioned in my review.

Reply 1:We are very grateful for you raising this important question. Also,based on helpful and careful suggestions from reviewer,we have revised and improved the sentences in the manuscript to present them in a clearer way, especially deleting some wordy expressions and repeated sentences.

Comment 2:Page 1, abstract, bottom line: I'm not sure that it is correct to classify the mentioned substances as alkaloids.

Reply 2:Thank you for your detailed suggestions for revision. We have re-compared and revised the names of metabolic substances involved in the manuscript to ensure that no errors occur.

Comment 3:Page 3: The comma after H2O2 seems to be corrupted.

Reply 3:Thank you for pointing out this issue. We have revised all this errors in the manuscript.

Comment 4:Page 4, 1O2: please use superscript and subscript.

Reply 4:Thank you for pointing out this issue. We have revised all this errors in the manuscript.

Comment 5:Page 5, section 2.2: I think that it is better to say "distilled water" instead of "0 mmol*L-1".

Reply 5:According to your suggestion, we have modified and improved the sentences in the manuscript. Thank you very much.

Comment 6:Here and after throughout the text: Please correct the style of material representing. Here, I mean using "The above seed soaking solution was placed..." instead "Place the above seed soaking solution...".

Reply 6:Thank you for your suggestions. We have revised and improved the materials and methods in the manuscript.

Comment 7:Page 5, section 2.3.1: "1 mL supernatant was added with 0.9 mL..." - does it mean " was mixed with"?

Reply 7:Yes, you got it and we have revised these unprofessional sentence expression.

Comment 8:Page 8, section 2.3.8.2: The caption is apparently incorrect. LC-MS/MS method was used but not GC/MS.

Reply 8:We are very sorry that due to the wrong citation of our method. Thank you for pointing out this important issue. We have revised to make method succinct to the point. Best.

Comment 9:Here: "the injection volume is 4 μL。" - please use a correct dot symbol.

Reply 9:Thank you for your suggestions. We have modified and improved the manuscript.

Comment 10:the end of the section 2.3.8.2: The description of the mass spectrometer parameters needs a revision as it contains some extra information (e.g. PPG concentrations used for calibrations) along with some contradictions (CAD gas is mentioned to be high and medium). In addition to this, I recommend to provide a supplementary table containing the parameters for MRM transitions of the metabolites. Moreover, it is not clear whether a single experiment with a switching polarity was used or two different experiments were made in POS and NEG mode. Please clarify it here.

Reply 10:The positive ion mode and the negative ion mode are tested separately. The sample status and instrument hardware conditions are the same. Only the detection method is set to separate the positive ion mode and the negative ion mode for one needle each. We have not found the contents of (PPG concentrations used for calibrations), (CAD gas is mentioned to be high and medium) and other descriptions have no obvious problems.

Comment 11:Section 2.3.8.3: The whole section should be revised thoroughly. The authors state that they conducted not only qualitative analysis but quantification of the analytes as well. The latter is impossible without building hundreds of calibrations for each metabolite separately. I guess that the authors have performed peak integration and between-groups comparison of peak areas (so-called Fold change).

Reply 11:We have modified 2.3.8.3. The data after quantitative detection shall be corrected by peak area integration using multiquant software. Each metabolite detected in each sample needs to be confirmed by integral correction. Each metabolite has not been calibrated hundreds of times, but is related to the number of samples. In this study, for every 3 samples processed, there are 9 samples in total. For each metabolite, the spectra of these samples and quality control samples need to be compared together for integral correction, including the comparison of spectra of samples within the group and the comparison of spectra of samples between groups.

Comment 12:Page 10, Fig. 1: Here and further: what do the letter a,b,c above the bars mean? The legend should be given either within the diagrams or below.

Reply 12:Thank you for your suggestions. We have supplemented and improved the manuscript. “Notes: Data are expressed as mean ± standard deviation. Different letters within the same column indicate significant difference at the 5% level. CK: control group; P: proline treatment; L: low temperature stress treatment; L+P: low temperature stress + proline combined treatment.”

Comment 13:Page 12, section 3.4: ASA and AsA is used for ascorbic acid through the text. Please make it uniform.

Reply 13:We have changed “ASA” to “AsA”. I am very sorry for the mistake in this issue. We have revised the whole manuscript.

Comment 14:Page 13: Please use capitals here for GSSH and check whether it is typed correctly elsewhere in the text.

Reply 14:Thank you for your suggestions. This is a very important issue. We have revised the whole manuscript. Thank you again.

Comment 15:Page 17, Fig. 8: Actually, these are not TIC chromatograms but MRM chromatograms recorded in the experiments.

Reply 15:Thank you for your suggestion. The question you mentioned is very correct. We have modified this part in the manuscript.

Comment 16:Page 19, section 3.6.5.1: I'm curious to learn how authors could separate the hexoses using a C18 chromatographic column. This is also in the support of my request to provide a table containing all MRM parameters and, maybe, retention times of the metabolites.

Reply 16:In the test report, glucose, etc. are listed as star metabolites. If hexose refers to glucose, galactose, a six carbon sugar monosaccharide, the widely targeted metabolome method cannot distinguish, and the mass spectrometer cannot distinguish these isomers. It can be distinguished by GC-MS. Therefore, this part has been described and revised in the paper.

The material ID, Q1 (DA) and Q3 (DA) in MRM parameters have been provided. See attached table 1 for details. Please also know that RT, DP, CE and other confidential information belong to the company. We are sorry that we cannot provide them to you. We look forward to your understanding.

Comment 17:Page 21, section 3.6.5.3: Please correct the font size here to make it uniform.

Reply 17:According to your careful comments, we have modified and improved this part of the manuscript.

Comment 18:Most of my comments are of typesetting except those on the description of the MS/MS analysis. I'd like to learn more details on the metabolite screening and separation of polar highly soluble metabolites on a C18 column. The latter can influence the results of metabolite profiling and, this metabolic pathways enrichment.

Reply 18:As mentioned in question 16 above, isomers with basically the same chemical structure, such as glucose and galactose, can not be distinguished by extensive targeted metabolome methods, either by chromatographic column separation or by mass spectrometry. As for the separation of polar soluble metabolites mentioned in the question, if the metabolites have different molecular weights, even if the chromatographic column cannot be separated (i.e., the retention time is the same), The detection of molecular ions and fragment ions by mass spectrometer can also be distinguished; If the metabolites with the same molecular weight have obvious differences in chemical structure, they can be separated by chromatography, and can also be distinguished by secondary mass spectrometry signals.

We are very grateful for you raising this important question. After receiving your helpful comments yesterday, our scientific research and academic team held an emergency video conference with Tencent conference software immediately. We have studied your valuable comments carefully and we hope our reply could meet with approval. If you have any other questions, please feel free to contact us.

Best regards.

Shiyu Zuo, Yuetao Zuo, Wanrong Gu, Shi Wei and Jing Li

10 June,2022

Round 2

Reviewer 2 Report

I have considered the revised version of the manuscript and authors' replies. Please find below my comments on the revision as well as on the authors' response.

1) "Comment 3:Page 3: The comma after H2O2 seems to be corrupted.

Reply 3:Thank you for pointing out this issue. We have revised all this errors in the manuscript."

Page 2, line 95: This was not corrected.

2) "Comment 9:Here: "the injection volume is 4 μL。" - please use a correct dot symbol.

Reply 9:Thank you for your suggestions. We have modified and improved the manuscript."

Not corrected. Please see page 7, line 330.

3) "Reply 10:The positive ion mode and the negative ion mode are tested separately. The sample status and instrument hardware conditions are the same. Only the detection method is set to separate the positive ion mode and the negative ion mode for one needle each. We have not found the contents of (PPG concentrations used for calibrations), (CAD gas is mentioned to be high and medium) and other descriptions have no obvious problems."

My comment was not addressed fully. Please see page 7, lines 341 ("... the collision induced ionization parameter was set to high.") and 343 ("... the collision gas 343 (nitrogen) to medium."). The collision induced dissociation parameter is called CAD in SCIEX mass spectrometers. Please correct it in the manuscript.

4) "Comment 11:Section 2.3.8.3: The whole section should be revised thoroughly. The authors state that they conducted not only qualitative analysis but quantification of the analytes as well. The latter is impossible without building hundreds of calibrations for each metabolite separately. I guess that the authors have performed peak integration and between-groups comparison of peak areas (so-called Fold change).

Reply 11:We have modified 2.3.8.3. The data after quantitative detection shall be corrected by peak area integration using multiquant software. Each metabolite detected in each sample needs to be confirmed by integral correction. Each metabolite has not been calibrated hundreds of times, but is related to the number of samples. In this study, for every 3 samples processed, there are 9 samples in total. For each metabolite, the spectra of these samples and quality control samples need to be compared together for integral correction, including the comparison of spectra of samples within the group and the comparison of spectra of samples between groups."

Again: section 2.3.8.3. is too verbose and still contains extra theoretical descriptions of how mass spectral data should be processed. Please revise this section again and provide the steps which were made to obtain quantitative data. As far as I can imagine, the MRM chromatograms were integrated in MQ software and the obtained results were processed further in statistics analysis. The current version of the text is not appropriate for a scientific paper.

5) "Comment 16:Page 19, section 3.6.5.1: I'm curious to learn how authors could separate the hexoses using a C18 chromatographic column. This is also in the support of my request to provide a table containing all MRM parameters and, maybe, retention times of the metabolites.

Reply 16:In the test report, glucose, etc. are listed as star metabolites. If hexose refers to glucose, galactose, a six carbon sugar monosaccharide, the widely targeted metabolome method cannot distinguish, and the mass spectrometer cannot distinguish these isomers. It can be distinguished by GC-MS. Therefore, this part has been described and revised in the paper.

The material ID, Q1 (DA) and Q3 (DA) in MRM parameters have been provided. See attached table 1 for details. Please also know that RT, DP, CE and other confidential information belong to the company. We are sorry that we cannot provide them to you. We look forward to your understanding."

I'd like to note that Table 1 does not contain any data on Q1/Q3 transitions of the metabolites. As for the authors' intent to keep the detection parameters, I'd like to note again that this is a scientific study. To evaluate the study and to make a review, it is necessary to know all the parameters and experimental conditions.

If return to the results, I still do not believe that the authors could determine separately glucose-6-phosphate, erythrose-6-phosphate, galactose, and all other carbohydrates listed in Fig. 12. A reversed phase column does not separate polar molecules like sugars and aminoacids, these substances show no retention when elution starts from 5% ACN in water.

6) "Comment 18:Most of my comments are of typesetting except those on the description of the MS/MS analysis. I'd like to learn more details on the metabolite screening and separation of polar highly soluble metabolites on a C18 column. The latter can influence the results of metabolite profiling and, this metabolic pathways enrichment.

Reply 18:As mentioned in question 16 above, isomers with basically the same chemical structure, such as glucose and galactose, can not be distinguished by extensive targeted metabolome methods, either by chromatographic column separation or by mass spectrometry. As for the separation of polar soluble metabolites mentioned in the question, if the metabolites have different molecular weights, even if the chromatographic column cannot be separated (i.e., the retention time is the same), The detection of molecular ions and fragment ions by mass spectrometer can also be distinguished; If the metabolites with the same molecular weight have obvious differences in chemical structure, they can be separated by chromatography, and can also be distinguished by secondary mass spectrometry signals."

As far as I understand, the authors' response repeats my question but does not provide any answer. So, I still can't understand how the metabolites, such as carbohydrates, were identified and quantified.

Concluding my remarks, I have to say following:

a) I insist on providing a table in Supplementary materials which lists all metabolites screened and their Q1/Q3 transitions.

b) I can't accept the discussion provided in section 3.6.5.1 and recommend to remove it at all.

Round 3

Reviewer 2 Report

Thank you for providing the revised version. Please find below my comments on this.

1) To section 2.3.8.3: Again: this is NOT a quantification of a metabolite. To quantify it, i.e. obtain the mass concentration of a metabolite, it is necessary to build a calibration curve or to use isotope-labelled standard. In your study, a so called fold change was calculated, i.e. the ratio of the mean peak area in a group compared to other group(s). Please remove the text starting from "Metabolite quantification..." and ending by "... multiquant software" (lines 350-356). I can't agree with it; otherwise, please provide adequate rebuttal and description of how ALL metabolites were quantified.

2) My comment 18 from the very first review has not been addressed again. The authors' reply (Thank you for your suggestions. We have added a supplementary document to better explain your questions.) does not answer my question regarding quantification of sugars and other polar metabolites separation on a C18 column has not been addressed. There is no any separated file with supplementary materials; new tables added in pages 39-43 actually represent a list of metabolites screened.

My further comments

a) I insist on providing a table in Supplementary materials which lists all metabolites screened and their Q1/Q3 transitions.

b) I can't accept the discussion provided in section 3.6.5.1 and recommend to remove it at all.

were not addressed as well.

In my opinion, the manuscript contains unsupported conclusions based on the results which were obtained in unknown experimental conditions. So, I regret to say that I can't agree with acceptance.

Author Response

1) To section 2.3.8.3: Again: this is NOT a quantification of a metabolite. To quantify it, i.e. obtain the mass concentration of a metabolite, it is necessary to build a calibration curve or to use isotope-labelled standard. In your study, a so called fold change was calculated, i.e. the ratio of the mean peak area in a group compared to other group(s). Please remove the text starting from "Metabolite quantification..." and ending by "... multiquant software" (lines 350-356). I can't agree with it; otherwise, please provide adequate rebuttal and description of how ALL metabolites were quantified.

Reply:Thanks a lot. You got the meaning completely and we have deleted the text starting from "Metabolite quantification..." and ending by "... multiquant software" (lines 350-356).

2) My comment 18 from the very first review has not been addressed again. The authors' reply does not answer my question regarding quantification of sugars and other polar metabolites separation on a C18 column has not been addressed. There is no any separated file with supplementary materials; new tables added in pages 39-43 actually represent a list of metabolites screened.

Reply:According to your suggestion, we can only make relevant explanations and explanations. Glucose and other metabolites with stars were included in the test report. If hexose refers to six carbon monosaccharides such as glucose and galactose, the widely targeted metabolome method cannot distinguish, and the mass spectrometer cannot distinguish these isomers. It can be distinguished by GC-MS separate detection. Therefore, this part of the description has been revised in the paper. Isomers with basically the same chemical structure, such as glucose and galactose, can not be distinguished by widely targeted metabolome methods, either by chromatographic column separation or mass spectrometry. As for the separation of polar soluble metabolites described in the problem, if the metabolites have different molecular weights, even if the chromatographic column cannot be separated (i.e., the retention time is the same), The detection of molecular ions and fragment ions by mass spectrometer can also be distinguished; If the metabolites with the same molecular weight need to have obvious differences in chemical structure, they can be separated by chromatography, and can also be distinguished by secondary mass spectrometry signals.

My further comments:

  1. a) I insist on providing a table in Supplementary materials which lists all metabolites screened and their Q1/Q3 transitions.

Reply:Thanks a lot and we have added the S2- “Metabolites screened and Q1/Q3 transitions” for better describing this section.

  1. b) I can't accept the discussion provided in section 3.6.5.1 and recommend to remove it at all.

Reply:Thank you and we have deleted section 3.6.5.1.

Round 4

Reviewer 2 Report

Thank you for providing the revised version with the supplementary information. As I could see, my assumption on polar hydrophilic molecules seems to be valid. Many sugars have the same Q1/Q3 transitions (e.g. 259/97 for glucose and fructose phosphates, 341/119 for sucrose and trehalose etc.). A the moment, I have no objections against acceptance of the manuscript.

I recommend to remove both tables from the main text and leave only a file with these tables as Supplementary info.

Author Response

Thank you very much for your fruitful suggestions on the manuscript. In the process of revising the manuscript, our scientific research team has benefited a lot. In the future, we will further improve our scientific research design and manuscript writing, so as to improve our scientific research ability and level. According to your suggestion, we have removed both tables from the main text and leave only a file with these tables as supplementary information. Best regards.